

# Decadal resolution record of Oman margin upwelling indicates persistent solar forcing of the Indian summer monsoon after the early Holocene summer insolation maximum

Philipp M. Munz[1], Stephan Steinke[2], Anna Böll[3], Andreas Lückge[4], Jeroen Groeneveld[5], Michal Kucera[5], and Hartmut Schulz[1]

[1]Department of Geosciences, University of Tübingen, Hölderlinstr. 12, 72074 Tübingen, Germany
[2]Department of Geological Oceanography, Xiamen University, Xiping Building, Xiang'an South Road, Xiamen 361102, China
[3]Institute of Geology, University of Hamburg, Bundesstr. 55, 20146 Hamburg, Germany
[4]Bundesanstalt für Geowissenschaften und Rohstoffe (BGR), Stilleweg 2, 30655 Hannover, Germany
[5]MARUM – Center for Marine Environmental Sciences, Germany, Leobener Str., 28359 Bremen, Germany

*Correspondence to:* Philipp Munz (philipp.munz@uni-tuebingen.de)

**Abstract.** The Indian summer monsoon (ISM) brings most of the annual precipitation to the densely populated region in southern Asia. For the agricultural development and economic prosperity of the region, it is therefore vital to assess the variability of the monsoon system on societal relevant decadal- to centennial time scales. This might help to better understand how potential driving forces might be controlling ISM variability and how it might develop under future climate scenarios. Here we
5  present a study of a sediment core from the northern Oman margin, revealing early- to mid Holocene ISM conditions on a near 20-year resolution. We assess multiple independent proxies indicative of sea surface temperatures (SST) during the upwelling season together with bottom water conditions. We use geochemical parameters, transfer functions of planktic foraminiferal assemblages and Mg/Ca paleothermometry and find evidence corroborating previous studies that upwelling intensity varies significantly in coherence to solar sunspot cycles. The dominant ~80–90-year Gleissberg cycle was apparently also affecting
10  bottom water oxygen conditions. Although the interval from 8.4 to 5.8 ka B.P. is relatively short, the gradually decreasing trend of summer monsoon conditions was interrupted by short events of intensified ISM conditions. Results from both independent SST proxies are linked to phases of weaker OMZ conditions and enhanced carbonate preservation. This indicates that atmospheric forcing was intimately linked to bottom water properties and state of the OMZ on decadal time scales.

## 1 Introduction

15  The Indian summer monsoon (ISM) is the dominant driver for intraseasonal changes of wind directions and precipitation patterns in one of the world's most densely populated regions. To precisely determine possible changes under climatic conditions in response to anthropogenic impact, it is vital to understand how ISM variability is modulated on societal relevant decadal- to centennial time scales. In comparison to modern times, the early- to mid Holocene is marked by different orbital configurations, affecting Northern Hemisphere insolation (Berger, 1978a) and monsoon circulations (Kutzbach and Street-Perrott, 1985). This



period was marked by a steep decline of solar insolation changes whilst climatic conditions were largely unaffected by human induced greenhouse gas emissions. Furthermore, climatic conditions during the early- to mid-Holocene have been attributed to cultural and societal turnover in Africa (Kuper and Kröpelin, 2006) and Mesopotamia (Kennett and Kennett, 2007).

Monsoonal winds develop when summer (July to September) heating over the continent forms a low-pressure zone over continental Asia, which then interacts with the high-pressure zone over the southern Indian Ocean and drives strong, moisture-laden southwesterly winds. Along the eastern coast off Somalia and Oman, the alongshore winds induce coastal upwelling of cold and nutrient-rich deeper water layers, which cool surface water temperatures and fuel biological productivity within the euphotic zone (Findlater, 1969; Wyrtki, 1973). Below the photic zone, remineralisation of sinking organic matter consumes oxygen, which leads in combination with the lateral supply of low-oxygenated intermediate water masses (You and Tomczak, 1993), to the formation of a pronounced mid-depth oxygen minimum zone (OMZ). Upwelling and OMZ intensity, together with the biological uptake/release of carbon dioxide ($CO_2$), nitrous oxide ($N_2O$) and other greenhouse gases from/into the atmosphere (Farías et al., 2009; Paulmier et al., 2011; Ward et al., 2009), play a critical role for the global climate system.

Quantitative reconstructions of upwelling intensity have the advantage of being directly related to wind strength and thus ISM intensity (Murtugudde et al., 2007). Numerous studies attempted to quantitatively reconstruct sea surface temperatures (SST) in the northwestern Arabian Sea to study changes in upwelling and thus ISM intensity (Emeis et al., 1995; Clemens et al., 1991; Naidu and Malmgren, 1996; Dahl and Oppo, 2006; Huguet et al., 2006; Anand et al., 2008; Godad et al., 2011). Accordingly, ISM variations are modulated by Northern Hemisphere summer insolation changes controlled by orbital parameters of the earths' precessional cycle. However, previous reconstructions did not resolve SST variations on societal relevant decadal- to centennial timescales. Here, we present a high-resolution study of early- to mid Holocene ISM variability based on a multi-proxy study comprising of SST fluctuations and OMZ intensity. Reconstructions are based on a comparison of Mg/Ca measurements of the upwelling-related species *Globigerina bulloides* and on transfer functions based on planktic foraminiferal (PF) census counts. To this end we first evaluate species response to potential ecological driving parameters and convert fossil assemblage counts to summer SST using a regionally validated transfer function (Munz et al., 2015). We test the ecological importance and statistical significance of the reconstructions and evaluate summer SST variability in comparison to other proxy records. We further assess the influence of surface parameters, upwelling SST and primary productivity, on variations of OMZ intensity and deep-water carbonate preservation.

## 2   Modern oceanographic setting

Modern hydrographic conditions in the Arabian Sea are strongly dominated by the seasonal monsoon cycle. During boreal summer (June–September), strong southwestern monsoonal winds drive surface currents in the western Arabian Sea (Figure 1), namely the Somali Current (SC) and as its northward extension the East Arabian Current (EAC), promoting an overall anti-cyclonic circulation pattern (Shetye et al., 1994; Tomczak and Godfrey, 1994). The alongshore surface currents induce an offshore-directed deflection of surface waters and upwelling of deeper, cold and nutrient-rich subsurface waters through the process of Ekman-pumping. During boreal winter (December–March), weaker, dry and cold northeasterly winds (IWM; Indian



winter monsoon) lead to increased surface water cooling and evaporation, especially in the northeastern part of the basin, where it enables deep convective mixing and a second productivity peak during winter (Madhupratap et al., 1996; Schulz et al., 2002; Munz et al., 2015, 2016).

Enhanced nutrient availability during summer upwelling nourishes primary productivity within the photic zone near the Somali- and Omani coast. This promotes OMZ conditions at intermediate water levels (~100–1000 m; Figure 1). The water masses in the Arabian Sea are derived from several major sources. High evaporation leads to highly saline surface water, which is therefore named Arabian Sea High Salinity Water — ASHSW (Shetye et al., 1994; Prasanna Kumar and Prasad, 1999). Thermocline waters are mainly sourced by an intermediate water mass (500–1500 m water depth) that originates in the central Indian Ocean (Indian Ocean Central Water, IOCW) as a mixture of Antarctic Intermediate Water (AAIW) and Indonesian Intermediate Water (IIW) (Emery and Meincke, 1986; You, 1998). Although these waters have a relatively low pre-formed oxygen saturation when they arrive in the Arabian Sea, the Subantarctic Mode Water (SAMW) and AAIW exert a strong influence on the OMZ of the Arabian Sea (Böning and Bard, 2009). Additionally, intermediate waters are sourced by the inflow of highly saline waters forming through excess evaporation over precipitation in the Persian Gulf and Red Sea. This leads to the formation of high salinity subsurface waters and subduction under less saline waters after entering the Arabian Sea. The Persian Gulf Water (PGW) shows a pronounced salinity maximum at 200–400 m water depth, whereas the Red Sea Water (RSW) has an equilibrium depth of 500 m at its maximum northward extent at ~18°N (Shetye et al., 1994). Although the PGW is a relatively young water mass and originates from a closer source region compared to the other intermediate water masses, the high-salinity tongue of modern PGW increases the oxygen concentration at mid-depth by less than 0.5 ml l$^{-1}$ (Figure 1).

## 3 Material and methods

### 3.1 Samples and chronology

During RV *METEOR* cruise M74 on Leg 1b in September 2007 (Bohrmann et al., 2010), piston core SL163 recovered the uppermost 955 cm of the sedimentary sequence from 650 m water depth within the central part of the present-day OMZ (Figure 1) at the northern Oman Margin (21°55.97'N; 059°48.15'E). Multicorer MC681 recovered the undisturbed uppermost 53 cm of near-surface sediments at the same position. The sedimentary sequence from the intervals 1–56 cm of core SL163 and 1–53 cm of MC681 is considerably different from the topmost 1 cm and the deeper intervals of SL163. From smear slide analysis and scanning electron microscopy this deposition can be described as faintly bioturbated, olive-brown organic-rich diatomaceous nannofossil silty clay. The uppermost 1 cm from both cores as well as the deposition below 56 cm of SL163 is described as olive foraminiferal nannofossil ooze. Core SL163 was sampled in continuous 1–2.5 cm intervals and for this study only samples from the foraminiferal nannofossil ooze between 56 cm and 400 cm of SL163 and the core top (0–1 cm) of MC681 were investigated.

The age model of core SL163 and the core top sample of MC681 is based on fourteen AMS $^{14}$C datings (Table 1), measured at Beta Analytics, Miami/USA and the Leibniz Laboratory in Kiel/Germany. Datings of core SL163 were mostly based on a monospecific analysis of handpicked individuals of *Neogloboquadrina dutertrei* (Table 1). This species is commonly known



to thrive at thermocline depths (e.g., Fairbanks et al., 1982), but in the western Arabian Sea it shows a relatively shallow habitat within the upper 35 m of the water column (Peeters and Brummer, 2002). For two samples (52.5 cm and 58.5 cm) with insufficient foraminiferal carbonate content, the bulk organic fraction was dated. To avoid a potential bias introduced by dating different organic compounds that might have been produced during different seasons or within different water masses, the core

top sample of MC681 was sampled twice using mixed planktic foraminifera and bulk organic fraction. Both analyses yielded comparable results (Table 1), indicating a similar synthesis and general comparability of both dating methods.

Radiocarbon dates were calibrated to calendar years using the MARINE13 calibration curve (Reimer et al., 2013) within the program clam ver. 2.2 (Blaauw, 2010) for the statistical software environment $R$ ver. 3.2.1 (R Core Team, 2015). We assume a regional reservoir correction of $\Delta R = 231 \pm 31$ years, which is the weighted mean of the four closest (max. distance ~700 km)

$\Delta R$ determination points from the 14CHRONO Database (http://calib.qub.ac.uk/marine/). Added to the global marine reservoir age of 400 years, this correction factor is consistent with the recent age of our undisturbed core top sample of MC681 (Table 1). An age-depth model was then built for SL163 using a smooth spline regression with a mixed effect model (10,000 iterations) within clam.

### 3.2 Planktic foraminiferal faunal analyses and transfer function approach

Samples were freeze-dried, washed with tap water over a 63 µm screen, oven-dried at $40\,^{\circ}\mathrm{C}$ and dry-sieved into the size fractions 150–250 µm, 250–315 µm and >315 µm. Planktic foraminifera (PF) were identified according to the taxonomic framework of Hemleben et al. (1989); Bé (1967); Bé and Hutson (1977). Following the procedure of Pflaumann et al. (1996), PF counts were conducted on a sub-split of each size fraction that contained a minimum of 100 PF individuals, yielding a summed fraction (>150 µm) to contain >300 identified PF individuals. Relative species abundances were calculated on the

summed fraction >150 µm after multiplying each size fraction with the respective split factor. To evaluate the secondary influence of calcite dissolution we calculated fragmentation indices following Le and Shackleton (1992).

Quantitative reconstructions from PF faunal abundances were based on a modern calibration dataset of $n = 603$ surface samples spanning the Indian Ocean from $30\,^{\circ}\mathrm{S}$ to $25\,^{\circ}\mathrm{N}$ and $33\,^{\circ}\mathrm{E}$ to $119\,^{\circ}\mathrm{E}$ recently compiled by Munz et al. (2015). To enhance the signal-to-noise ratio, rare species <0.5 % average abundance were removed from the calibration dataset prior

to further analyses (Kucera et al., 2005), which resulted in the exclusion of 16 rare species out of 33 taxa. The dataset was then further subsampled to contain only samples most similar with the fossil assemblages of core SL163. This removes noisy effects introduced by modern samples covering ecological conditions outside the range of the fossil samples (Kucera et al., 2005). We followed the approach presented in Munz et al. (2015) and restricted the calibration dataset to samples that had the same range along the first and second axis of a joint principal component analysis together with the fossil dataset (using

square-root transformed species abundances to dampen the effect of a few dominant taxa). The resulting subsampled dataset used for further analyses contained $n = 96$ modern surface samples (Figure 2).

Environmental gradient analysis was then carried out on the 96 subsampled calibration samples using the package vegan ver 2.3-1 (Oksanen et al., 2015) for $R$. An initial detrended correspondence analysis (DCA) revealed a short gradient length (0.75 for the longest gradient on DCA axis 1), suggesting an ordination method based on linear species response (Birks, 1998).



Redundancy analysis (RDA) was therefore chosen for the construction of an ordination model, to examine species response to several potential controlling environmental gradients. Because of the extreme seasonality of surface water productivity owed to the monsoonal circulation, four different approaches were used to assess modern productivity values. Net primary productivity estimates from the Vertically Generalized Production Model (VGPM), the Eppley-VGPM and the Carbon-based Productivity

Model (CbPM) were accessed through the Ocean Productivity Home Page (http://www.science.oregonstate.edu/ocean.productivity/). Satellite-derived chlorophyll $\alpha$ measurements of SeaWiFS and Aqua MODIS sensors averaged over the period 1998–2010 were obtained from NASA Ocean Biology (http://oceancolor.gsfc.nasa.gov/cms/). Sea surface salinity (SSS) and sea surface temperature (SST) were extracted from the 10 m depth level of the World Ocean Atlas 1998 (Conkright et al., 1998).

Transfer function analysis and statistical significance tests of the reconstructions were performed with the $R$ packages rioja

ver 0.9-5 (Juggins, 2015) and palaeoSig ver. 1.1-3 (Telford, 2015). We used two common techniques for paleoenvironmental reconstructions, the Imbrie-and-Kipp method — IK (Imbrie and Kipp, 1971) and weighted averaging partial least squares regression — WA-PLS (ter Braak and Juggins, 1993) to reconstruct SST from the fossil samples. The number of factors to extract for IK was based on the Kaiser-Guttman criterion, the Parallel Analysis after Horn (1965) and the analysis of optimal coordinates (reviewed by Courtney and Gordon, 2013). WA-PLS model complexity was evaluated from the model of lowest

root mean squared error of prediction (RMSEP) using bootstrapping cross-validation ($n = 999$ cycles).

### 3.3 Mg/Ca analyses and paleothermometry of *G. bulloides*

For Mg/Ca analyses approximately 30 individuals of *G. bulloides* were picked from the size fraction 250–315 μm, gently crushed between two glass plates under the microscope to open the chambers and cleaned following the protocol of Barker et al. (2003). Trace elemental measurements were performed with an inductively coupled plasma optical emission spectrometer

(ICP-OES) using an Agilent Technologies 700 Series with autosampler ASX-520 Cetac and micro-nebulizer at the MARUM-Center for Marine Environmental Sciences, University of Bremen/Germany. Precision of ICP-OES measurements was determined using an in-house standard with a Mg/Ca ratio of 2.93 mmol mol$^{-1}$ after every five samples. The average relative standard deviation from $n = 55$ measurements was 0.155 % ($1\sigma = 0.005$). As a standard reference material, we analysed the international limestone standard ECRM752-1 with an Mg/Ca ratio of 3.75 mmol mol$^{-1}$ (Greaves et al., 2008) prior to every batch

of 50 samples. Average Mg/Ca values over all ECRM752-1 measurements was 3.714 ($n = 8$; $1\sigma = 0.041$). Three replicate measurements of every sample were used to estimate analytical precision and yielded an average relative standard deviation of 0.068 % ($n = 234$ samples, $1\sigma = 0.004$). To estimate SST from Mg/Ca we used the species-specific calibration equation $T = 1/0.102 \times LN(\text{Mg/Ca}/0.528)$ published by Elderfield and Ganssen (2000) for *G. bulloides*. This Mg/Ca-temperature relationship was previously used for calibrating Mg/Ca measurements of *G. bulloides* in the Arabian Sea (Anand et al., 2008;

Ganssen et al., 2011).

Samples were screened for a potential contamination by iron-manganese coatings and clay minerals not successfully removed by the cleaning technique (Barker et al., 2003). Ten out of 142 samples showed Fe/Ca or Mn/Ca values of more than 0.1 mmol mol$^{-1}$ and were excluded prior to further analyses. Paleotemperature estimates based on trace elemental concentrations in foraminiferal calcite can potentially suffer from post-depositional dissolution and preferential removal of more solution



susceptible Mg-rich calcite (e.g., Brown and Elderfield, 1996; Dekens et al., 2002). We therefore tested a systematic dissolution bias of samples from core SL163 using a cross correlation of Mg/Ca ratios and the fragmentation index of Le and Shackleton (1992). If Mg/Ca measurements were affected by carbonate dissolution, a strong negative relationship between Mg/Ca ratios and fragmentation indices would be expected, which is not the case for our samples ($r = -0.07$, $p = 0.43$). Furthermore, the

presence of pteropods indicates that selective dissolution did not affect Mg/Ca-ratios.

### 3.4   Bulk geochemical analyses

Bulk geochemical analyses of the sediment were performed on average every 5 cm with X-ray fluorescence (XRF). Concentrations of manganese (Mn) and vanadium (V) were quantitatively analysed as an indicator for bottom water redox conditions and state of the OMZ. After fusion of the samples with lithium metaborate at $1200\,°C$ for 20 minutes (sample/LiBO$_2$ = 1/5)

samples were measured using Philips PW 2400 and PW 1480 wavelength dispersive spectrometers at the Federal Institute for Geosciences and Natural Resources, Hannover/Germany. Instrumental precision of the results was controlled with certified reference materials (CRM) (i.e., BCR, Community Bureau of Reference, Brussels). The precision for major elements was generally better than $\pm 0.5$ % and better than 5 % for trace elements.

Biogenic opal was determined photometrically after wet alkaline extraction of biogenic silica (BSi) using a modification of

the DeMaster method (DeMaster, 1981). About 30 g dry sediment per sample was digested in 40 mL of 1 % sodium carbonate solution (Na$_2$CO$_3$) in a shaking bath at $85\,°C$. After treatment with 0.021 M HCl, the neutralized supernatant was analyzed after 3, 4 and 5 hours and the amount of BSi was estimated from the linear intercept through the time course aliquots. Slope correction was used to prevent an overestimation of BSi by dissolution of clay minerals at low BSi concentrations (Conley, 1998). Biogenic opal was then determined by multiplying the BSi concentrations with a factor of 2.4. Duplicate measurements

revealed a mean standard deviation of 0.13 %.

### 3.5   Spectral analyses

Spectral analyses on the proxy records from planktic foraminiferal transfer functions, Mg/Ca-SST and OMZ intensity with the multi-taper method (MTM; Mann and Lees, 1996) were computed with the SpectraWorks software kSpectra©ver. 3.4.5 and a red noise null hypothesis (Ghil et al., 2002) using the default setting of $p = 2$ and $K = 3$ tapers. A Gaussian band-pass filter was

used to reveal the signature of the dominant cycles in the data. After resampling the time series to the average sampling rate, filtering was carried out using the software program AnalySeries ver. 2.0.8 (Paillard et al., 1996). A cross wavelet transform (XWT; Grindsted et al., 2004) of both temperature proxy time series was calculated with the biwavelet package ver. 0.17.10 for $R$. MTM and XWT analyses were conducted on trend-removed time series interpolated to regular average sample spacings using piecewise cubic polynomial interpolation (function 'pchip' of the signal package ver. 0.7-6 for $R$). To estimate a linear

relationship of the low-frequency signals between the differently spaced time series, a new common time axis was produced where signals were consecutively averaged into 60-year long bins with a 20-year overlap.



## 4   Results

### 4.1   Age control

The 1–2.5 cm sample spacing yielded an average temporal sampling distance of ~19 years over the entire interval. We observed two samples where the age-depth relationship is reversed within the lower half of the core (Figure 3). However, the maximum age deviation is lower than the $2\sigma$ probability of both dating points, enabling to fit a smooth spline model with continuous deposition rates and continuously increasing ages. The sharp lithofacies change at 56 cm core depth is marked by a sedimentation hiatus of ~3600 years (Table 1). Based on the accumulation rates above and below the unconformity, this corresponds to a thickness of the missing sedimentary sequence of ~1.5 m. One possible reason for this might be that the high water content of the organic-rich diatomaceous nannofossil silty clay deposited above the foraminiferal nannofossil ooze led to gravitational instability at the steeply inclined northern Oman margin. Further discussion of this issue will be presented elsewhere.

### 4.2   PF faunal analyses of core SL163 and paleothermometry

A total of 29 PF morphospecies were identified in core SL163, whereof six species showed a total average abundance of >5% (Figure 4). The overall PF fauna is dominated by *Globigerina bulloides* (39.2%), followed by *Globigerinita glutinata* (11.5%), *Globigerinoides ruber* (11.2%), *Globigerina falconensis* (9.2%), *Globigerinoides sacculifer* (8.3%) and *Globigerinella siphonifera* (7.4%). Core top studies (Bé and Hutson, 1977; Hutson and Prell, 1980; Prell and Curry, 1981), plankton tow casts (Peeters and Brummer, 2002) and sediment trap studies (Curry et al., 1992; Conan et al., 2002; Conan and Brummer, 2000; Mohan et al., 2006) in the Arabian Sea indicate that *G. bulloides* is the dominant species during upwelling season. Relative abundances of *G. bulloides* were used in a number of studies to express upwelling and ISM intensity (Naidu and Malmgren, 1996; Anderson et al., 2002; Gupta et al., 2005). The high numbers of *G. bulloides* throughout the studied interval of core SL163 suggests highly elevated primary productivity during the early- to mid Holocene summer upwelling at this station.

Modern surface water properties and plankton productivity at the northern Oman margin are dominated by the seasonal upwelling during boreal summer. Because PF assemblages are dominated by species produced during the upwelling season (Curry et al., 1992), we investigated the environmental control on the PF fauna during summer (July to September; J-A-S). RDA results indicate, that summer SST correlates best with the first RDA axis and is the strongest determinant in the explanation of PF assemblages among the investigated parameters (Table 2). Transfer functions were therefore calibrated to summer SST.

The three methods used for determining the factor numbers to retain for the IK transfer function approach suggested a number of $n = 4$ factors. For WA-PLS, a two component model showed best model performance ($r^2 = 0.65$) and lowest cross-validated error estimates ($\mathrm{RMSEP} = 0.95\,^\circ\mathrm{C}$). Performance estimates for both methods are given in Table 3. Statistical significance of the reconstructions was tested using a novel method of random forest reconstructions (Telford and Birks, 2011). The analysis shows that summer SST can be reconstructed from fossil PF assemblages of core SL163 with a high statistical significance (Table 3; Figure 5). Reconstructed summer SST from both techniques (IK and WA-PLS) show a high linearity ($r = 0.82$; $p < 2.2\mathrm{E}-16$), indicating a low model-specific bias (Kucera et al., 2005). The resulting consensus of reconstructed summer SST from the transfer function techniques ranges between $23.3\,^\circ\mathrm{C}$ and $25.8\,^\circ\mathrm{C}$ (Figure 6b). Compared to modern




summer (J-A-S) SST (10 m depth interval), which is $25.8\,^{\circ}$C at this station (Conkright et al., 1998), reconstructed early- to mid Holocene summer SST are thus $<2.5\,^{\circ}$C colder.

Measured Mg/Ca values range between 3.9 mmol mol$^{-1}$ and 7.3 mmol mol$^{-1}$ (Figure 4), corresponding to water temperatures ranging from $19.6\,^{\circ}$C to $25.7\,^{\circ}$C. The Mg/Ca value of 6.66 mmol mol$^{-1}$ of the modern core top sample at this station (MC681) yields SST estimates ($T = 24.85\,^{\circ}$C) close to modern summer SST of the upper 50 m of the water column (WOA1998 $=$ $24.81\,^{\circ}$C), indicating that the calibration equation of Elderfield and Ganssen (2000) is applicable to the Mg/Ca-temperature dependence of *G. bulloides* at the northern Oman margin. We are aware of a potential bias introduced by a genotype-specific fractionation of Mg/Ca ratios, which was recently shown to occur between a warm-water (type-I) and cool-water (type-II) preferring species of *G. bulloides* in the Arabian Sea (Sadekov et al., 2016). However, the high numbers of *G. bulloides* observed throughout the studied interval of core SL163 suggests intense upwelling conditions and we therefore assume that we mainly sampled specimens of the cool water genotype, Type IIf, that is restricted to upwelling stations of the northern Oman margin (Sadekov et al., 2016).

### 4.3 Proxies of OMZ conditions and carbonate preservation

Trace-element distributions of manganese (Mn) and vanadium (V) were used as an indicator for bottom water redox conditions and state of the OMZ, following (Tribovillard et al., 2006). The values were expressed as enrichment to average shale (Wedepohl, 1971). V is enriched throughout the studied interval and enrichment relative to average shale ranges from 1.0 to 1.7. With average values of 0.92, Mn is mostly depleted except for four short phases centered at 6.0, 6.3, 7.2 and 8.4 ka B.P. (Figure 6g). The concentration of pteropod fragments ranges from 0 (pteropod-barren) to $5.2x10^3$ numbers per gram dry weight and PF fragmentation ranges between 0.1 % and 12.2 % (Le and Shackleton, 1992).

## 5 Discussion

### 5.1 Interpretation of the two independent planktic foraminiferal SST proxies

A direct comparison of the Mg/Ca-SST of *G. bulloides* and assemblage-based (consensus of IK and WA-PLS) summer SST reveals no linear relationship ($r = 0.02$, $p = 0.91$), although the binned time series show a weak positive linearity ($r = 0.24$, $p < 0.06$). This indicates that the low frequency signals of both time series are linearly related. Furthermore, the time series of both proxy records reveal a common trend of increasing temperatures over the record (Figure 6a-b) and are coherent on a wide range of frequencies. Significant (>95% confidence) coherence was found on ~1300, ~110 and ~60 year periods, as well as on ~150, ~90 and ~40 year periods (>99% confidence). Both SST records are thus in good general agreement, but Mg/Ca SST are on average $2.3\,^{\circ}$C colder. This relative offset of the SST reconstructions might be attributed to timing differences during the recording of the respective proxy signal. *G. bulloides* is interpreted to carry the upwelling signal during peak production of this species (Peeters et al., 2002; Anand et al., 2008). Highest flux rates of *G. bulloides* are found from May to October, but with a clear maximum from late July to September (Curry et al., 1992; Conan and Brummer, 2000; Peeters et al., 2002).



Furthermore, several studies in the Arabian Sea (Peeters et al., 2002; Schiebel et al., 2004; Friedrich et al., 2012) and in the Java upwelling region (Mohtadi et al., 2011) indicate that *G. bulloides* thrives in coastal upwelling areas mostly within the mixed-layer and upper thermocline waters in the uppermost 50–60 m of the water column. Assemblage-based SST reconstructions are calibrated to the 10 m depth level of summer temperatures. We therefore interpret Mg/Ca values of *G. bulloides* to represent

calcification temperatures of mixed-layer and upper thermocline waters during peak production of the late summer upwelling and assemblage-based SST reconstructions to represent a shallower SST average from July to September.

### 5.2   Evidence for early- to mid Holocene monsoon variability

The range of reconstructed water temperatures of $6.1\,^{\circ}$C from Mg/Ca measurements of *G. bulloides* (upwelling SST) suggests that upwelling temperatures at the northern Oman Margin fluctuated strongly and rapidly during the relatively short interval

of the early- to mid Holocene covered by SL163. The amplitude of temperature fluctuations recorded by *G. bulloides* is approximately twice as high compared to oxygen isotope temperatures of the same species at the Somali Margin during the early Holocene (Jung et al., 2002). Records from the Arabian Sea revealed, that strongest ISM conditions occurred during the early Holocene around 8–11 ka B.P. (Staubwasser et al., 2002; Gupta et al., 2003; Thamban et al., 2007), followed by a gradual weakening around 7 ka B.P., which is concomitant with the time period of relatively warm assemblage-based SST. However, the

overall trend in our record is not consistent and rather indicates that gradual weakening of ISM conditions established as early as 8 ka B.P. Consistently low SST suggest upwelling and ISM intensity was strongest during the intervals ~7.5–8.1 ka, followed by events of strong ISM at ~7.0–7.3 and 6.1 ka B.P. (Figure 6a). Weaker ISM conditions are reflected by warmer-than-average SST values at 6, 6.5–6.9, 7.4 and 8.2 ka B.P. Stalagmite records from the nearby Hoti and Qunf caves (Figure 6i-j) indicate low monsoonal precipitation at 6.3, 7.4 and 8.3 ka BP (Fleitmann et al., 2007; Neff et al., 2001), suggesting that summer monsoonal

winds (reflected by low summer/upwelling SST) and amount of precipitation are in-phase during the early- to mid-Holocene. This is an interesting finding, as Fleitmann et al. (2004) show an apparent decoupling of ISM precipitation amount and wind intensity (represented by percentage of *G. bulloides*) since the year ~1900 AD, indicating that higher upwelling intensities do not necessarily involve increasing precipitation.

MTM analysis revealed statistically significant periodicities of assemblage-based summer SST at ~1300 and 75–95 years per

cycle (Figure 7a). Mg/Ca-based upwelling SST are modulated on frequencies at 110–130, 80–90 and ~40 years (Figure 7b). The longer ~110–130 year cycle was previously found in a number of records from the Asian monsoon realm (Berger and von Rad, 2002; Dykoski et al., 2005; Gupta et al., 2005) and is close to the 132-year sunspot cycle previously identified to be modulating the Oman upwelling system during the Holocene (Gupta et al., 2005). The ~80–90-year cycle has been observed in several studies of ISM variability (Neff et al., 2001; Fleitmann et al., 2003; Dykoski et al., 2005; Gupta et al., 2005) and

was interpreted to be most likely influenced by the 88-year solar Gleissberg cycle. We tested a possible solar component on the decadal-scale forcing of our SST records by evaluating the coherence of both time series with the record of reconstructed sunspot numbers (Solanki et al., 2004). The coherence pattern reveals, that both SST records and sunspot numbers are coherent on a wide range of periodicities (630, 190–230, 160, 110–130, 80–90, ~70, ~50 and ~40- years per cycle, Fig. 6d and e). This





observation further strengthens the hypothesis, that ISM variability is not only controlled by orbital-scale insolation forcing, indicated by the long-term trend of warming temperatures and decreasing ISM intensity, but also by solar forcing.

To illustrate the signature of the observed cycles in the data, band-pass filter outputs of both SST records are shown in Figure 8. Apparently, amplitude modulations of the filter outputs of both SST time series run largely synchronous. Comparison of the amplitude variations of the band-pass filtered assemblage-based summer SST time series with mean summer irradiance at $30°$N indicates that phases of largest amplitude fluctuations of summer SST are apparently lagging solar irradiance by ~200 years. However, regardless of the exact correlation of both signals, amplitude modulation of upwelling SST in the ~85- and ~120-year bandwidth are in-phase with amplitude modulation of solar irradiance. Our new high-resolution record of summer/upwelling SST therefore provides further strong evidence that solar forcing was persistently modulating ISM variability during the early- to mid Holocene after the summer insolation maximum.

### 5.3 Carbonate preservation and bottom water redox conditions

In order to study the interplay of ISM intensity and OMZ fluctuations, we compared our new multi-proxy reconstruction of summer SST to OMZ reconstructions of Böll (2014). This study previously evaluated ISM intensity and OMZ conditions during the deposition of SL163 in a spatial context using alkenone-derived annual mean SST, stable nitrogen isotopes and published OMZ reconstructions, concluding that changes of OMZ intensity were linked to variability of intermediate water ventilation and monsoon strength. Our new multi-proxy records primarily responding to summer upwelling conditions enables to study the interplay of ISM intensity and OMZ fluctuations during the early- to mid Holocene in more detail. Under oxic bottom-water conditions, Mn is precipitated as Mn oxy-hydroxides. Under suboxic or hypoxic bottom-water conditions, MnO can be reduced to soluble $Mn^{2+}$ and escape into the water column (Calvert and Pedersen, 1993; Calvert et al., 1996; Schnetger et al., 2000; Böning et al., 2004). Mn is mostly depleted in core SL163, indicating Mn-loss through the OMZ, except for three short intervals of Mn enrichment centered at ~6.0, 6.3 and 7.2 ka B.P., as well as from 8.3 ka B.P. to the end of the record (Figure 6g). However, precipitation and fixation of $Mn^{2+}$ in the sediment as $MnCO_3$ can occur when bottom-waters become completely anoxic (Tribovillard et al., 2006; McKay et al., 2015). This finding corroborates the study of Böll (2014), who found increased denitrification indicating more intense OMZ conditions during the intervals 5.9, 6.1, 6.4 and 7.2 ka B.P. (based on the same age model). V is precipitated under anoxic conditions (Emerson and Huested, 1991) and constant enrichment values between 1.0 and 1.7 relative to average shale also indicate permanent anoxic bottom water conditions during the deposition of core SL163.

Carbonate preservation was assessed by PF fragmentation indices and the abundance of pteropod fragments in the sieve fraction >150µm. Pteropod shells primarily consist of aragonite, the less stable polymorph of calcium carbonate as compared to calcite (Morse et al., 1980). In the northern Arabian Sea, aragonite has a modern compensation depth (ACD) of ~500 m (Berger, 1978b; Böning and Bard, 2009) and a lysocline at ~1 km water depth (Böning and Bard, 2009), thus approximately 350 m below the station of core SL163. However, in high-productivity environments supra-lysoclinal dissolution can occur well above this depth, induced by respiration of organic matter and metabolic release of $CO_2$ (e.g., Milliman et al., 1999). Thus, enhanced pteropod preservation indicates either ACD deepening (Reichart et al., 2002) or less supra-lysoclinal dissolution



due to decreased production of organic matter. Phases of increased concentrations of pteropod fragments are linked to the short intervals of slightly enriched Mn values (Figure 6f-g). This indicates short phases of less corrosive bottom waters due to potentially weaker OMZ conditions. Furthermore, the variability of Mn/Al values (expressed as Mn enrichment) is modulated on dominant ~170- and ~310-year cycles (Figure 7c), which is approximately half (170) and one fourth (340) of the ~80–90-

year frequency of the Gleissberg cycle. This suggests that bottom water oxygenation state is modulated on the same frequencies as SST variability and upwelling intensity.

The intervals of increased pteropod concentration and lower OMZ intensity at ~6, 6.3, 7.2 and 8.3–8.5 ka B.P. are linked to intervals of low biogenic opal and thus occurred when surface water productivity was diminished. The former three of these intervals also correspond to low abundances of the upwelling indicator species *G. bulloides* (Figure 6d), but during the latter

interval at 8.3–8.5 ka B.P. *G. bulloides* abundances are high (>45%). In addition, Mg/Ca measurements of *G. bulloides* at 8.3 ka B.P. are >7 mmol mol$^{-1}$ and inferred SST are >25.5°C. A similar feature of a very short warm excursion of Mg/Ca temperatures from *G. bulloides* with concomitantly high abundances of this species around 8 ka B.P. was observed by Anand et al. (2008) in their record from the Somali upwelling system. It may be speculated here whether a potential intrusion of the warm water genotype of *G. bulloides* (Sadekov et al., 2016) during the short-lived period around 8.3 ka B.P. had occurred,

leading to erroneously warm Mg/Ca temperature estimates during this period.

Regardless of the apparent difference of absolute temperatures, cross wavelet transform (XWT) reveals three distinct intervals, two in the ~40–90-year band (6–6.2, 7.2–7.6 ka B.P.) and one in the ~110–130-year band (8.1–8.4 ka B.P.), where both records show significant common power (Figure 9). The intervals correspond well to the phases of enhanced pteropod preservation and lower OMZ intensity. The cross wavelet phase angle of XWT further indicates that both SST records are

phase-shifted during these intervals. The arrows in Figure 9 within the significant areas are pointing mostly upwards, which indicates either that assemblage-based SST are leading Mg/Ca-SST by $\pi/2$, i.e. 90°, or that assemblage-based SST are lagging Mg/Ca-SST by $3\pi/2$, i.e. 270°. Although these phases are relatively short-lived and both scenarios seem reasonable from the visual comparison of the two SST time series, a systematic lead of faunal SST reconstructions compared to alkenone-derived SST was previously found by Cayre and Bard (1999) in a study from the eastern Arabian Sea. However, their study observed

a delay of several ka, which was potentially produced by strong productivity changes (Bard, 2001). It should be noted that the phase relationships at the longer wavelength could be erroneous, as the behaviour is not consistent and partly truncated by edge effects below the cone of influence.

Several studies in the Arabian Sea indicate an alternating influence of different water masses during the deglacial and early Holocene (Zahn and Pedersen, 1991; Jung et al., 2001; Schmiedl and Leuschner, 2005; Gupta et al., 2008; Jung et al., 2009).

The site of core SL163 is within the modern range of intermediate water masses, that form a mixture of RSW, PGW and IOCW down to ~1500 m water depth (Emery and Meincke, 1986; Shetye et al., 1994). During the last glacial stage, the outflow from the two mediterranean basins connected to the Arabian Sea (Red Sea and Persian Gulf) was highly suppressed due to lowering of the global sea level potentially close to or even below the respective sill depths (Rohling and Zachariasse, 1996). Gupta et al. (2008) discussed that ventilation changes at the northern Oman Margin during the early Holocene are distant to RSW and

more likely attributed to an alternating contribution of NADW to the deeper water masses. In comparison to their study site,

our core site is further north, close to the modern Ras-al-Hadd frontal zone, where PGW enters the Arabian Sea. At the station of SL163, modern PGW reaches down to 400 m (Figure 1). During phases of more restricted water exchange and increased evaporation, subsurface salinities in the Gulf might have been higher, which could have enabled PGW to reach the core site at deeper depth. However, the modern oxygen input of PGW is very low and OMZ conditions at this depth are still fully hypoxic.

Instead, the concomitant occurrence of high common cross wavelet power of the SST records (Figure 9) with less corrosive bottom waters indicates that deep water conditions are linked to SST variations and atmospheric forcing was intimately linked to the deep water properties.

## 6 Conclusions

Among the environmental variables investigated, summer SST is primarily controlling the underlying variability of PF assem-
blages from the NW Arabian Sea. Statistically significant reconstructions of summer SST from PF assemblages and Mg/Ca-SST from *G. bulloides* are colder than present summer temperatures, indicating vigorous monsoonal winds and upwelling intensity during the early- to mid Holocene. ISM conditions were generally strongest around 8 ka B.P. and gradually decreased towards 5.8 ka B.P. The trend of weakening monsoon conditions was interrupted by decadal-scale episodes of intensified ISM around ~7 and 6.2 ka B.P., which is concomitant to pluvial episodes indicated by the Oman speleothem records. Summer SST
fluctuations furthermore reveal, that upwelling intensity was coupled to decadal- to centennial-scale sunspot cycles.

Low monsoonal conditions, indicated by warm upwelling and summer SST, as well as low surface water productivity, are associated with enhanced pteropod preservation and weaker OMZ conditions. Enrichment of manganese as a proxy for bottom-water oxygenation varies on multiples of the dominant ~80–90-year Gleissberg cycle, suggesting that upwelling intensity and OMZ conditions are modulated by solar activity. We therefore conclude that, instead of intermediate water mass changes,
atmospheric forcing is the main driver of OMZ conditions.

## 7 Data availability

The data presented in this manuscript will be made available electronically at the PANGAEA Data Publisher for Earth & Environmental Science (www.pangaea.de).

*Competing interests.* The authors declare that they have no conflict of interest.

*Acknowledgements.* This study was conducted within the framework of the collaborative research project "CARIMA", funded by the German Ministry of Education and Research – BMBF (grant no. 03G0806C). We would like to thank Sofie Jehle and Dorothea Mosandl for lab assistance and sample preparation.



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



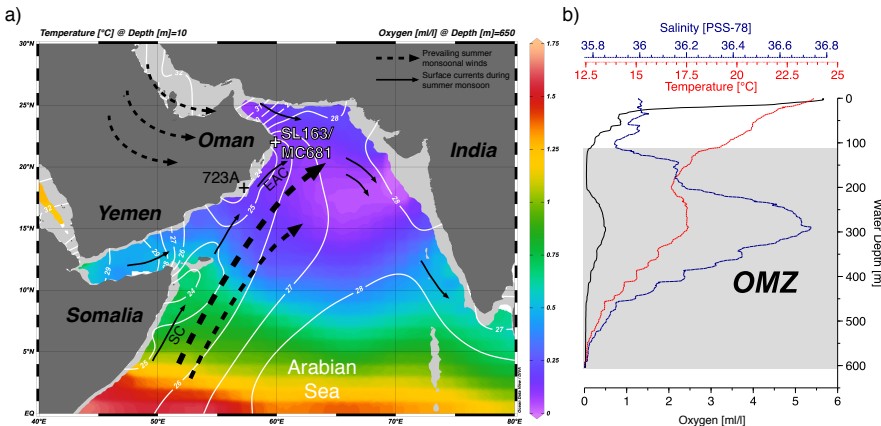

**Figure 1.** Map of the Arabian Sea, showing the location of gravity core SL163 and multicorer MC681 (white cross) at the northern Oman Margin (21°55.97'N; 059°48.15'E) in 650 m water depth, as well as ODP core 723A (black cross) studied by Gupta et al. (2005). Black arrows show the schematic pathways of major currents (solid lines; SC is Somali Current, EAC is East Arabian Current) and near-surface winds (stippled lines) during summer monsoon. White contour lines refer to the summer (July to September) sea surface temperature (SST) in 10 m water depth. Color shading indicates oxygen concentration (ml/l) in 650 m water depth, indicating a strong oxygen deficiency in the modern depth of the sampled sediment core. SST (Locarnini et al., 2010) and oxygen conc. (Garcia et al., 2010) are from World Ocean Atlas 2009. Inset figure shows the depth profile of the CTD data measured during cruise M74/1b at the station of core SL163. Grey shading refers to the location of the modern oxygen minimum zone (OMZ). A subsurface salinity maximum in 200–400 m water depth indicates the location of Persian Gulf Water (PGW). Map was created using the software ODV ver. 4.7.4 (Schlitzer, 2015).





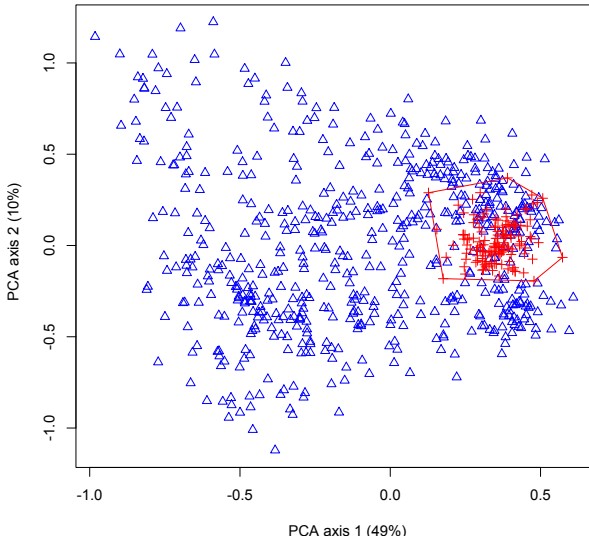

**Figure 2.** Scatterplot of the joint principal component analysis (PCA) of $n = 603$ modern calibration samples (blue triangles) and fossil downcore samples of SL163 (red crosses). The variance of the first and second PCA axis is indicated. $N = 96$ modern samples within the red polygon are most similar to the fossil samples along the first and second PCA axis, and were used for the calibration of transfer functions.




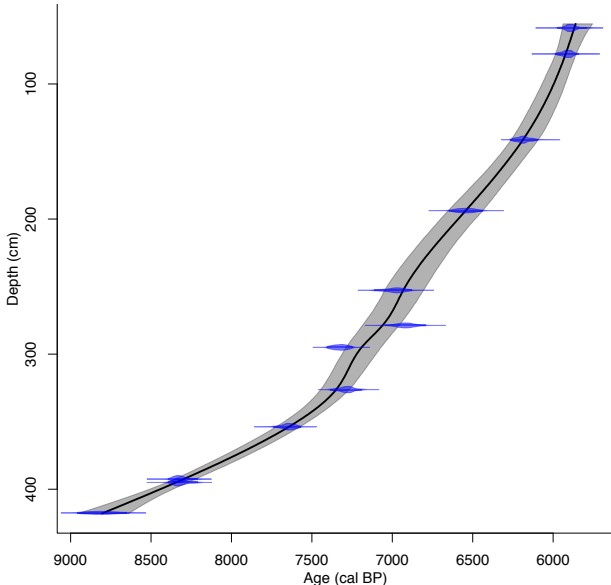

**Figure 3.** Age-depth relationship of the early- to mid Holocene section of core SL163 (55–400 cm). Blue areas indicate the conventional [14]C calibrated ages, the black line indicates the interpolation between the dated samples using a smooth spline fit and the 95 % confidence level (grey shading).



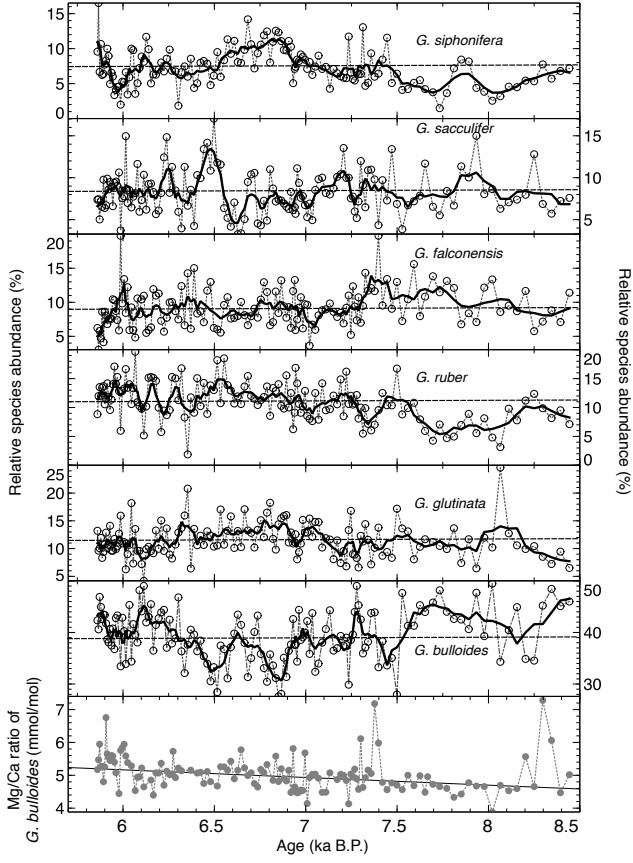

**Figure 4.** Time series of the six most common planktic foraminiferal species (>5% average abundance) and Mg/Ca measurements of *G. bulloides* from the studied interval in core SL163. Dashed lines are the real data, thick solid lines give the 3-pt running average.





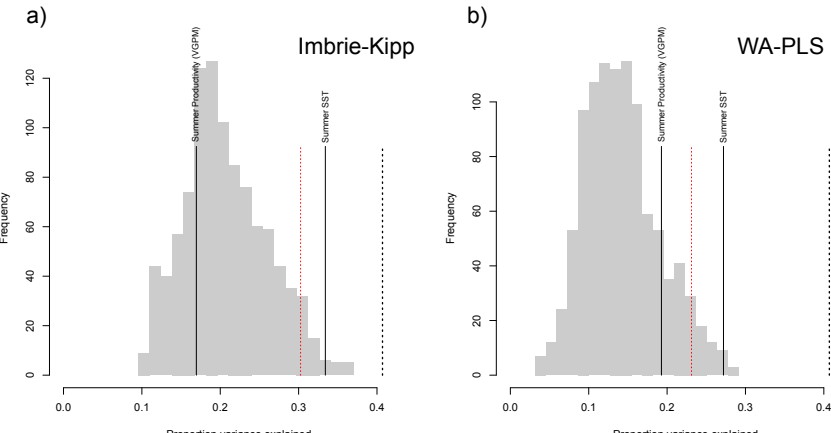

**Figure 5.** Results of the significance test of assemblage-based paleoenvironmental reconstructions using the R package palaeoSig (Telford, 2015). The null distribution is produced by generating 999 random environmental variables and reconstruct these variables from PF assemblages of core SL163 using **(a)** the Imbrie-and-Kipp method (IK) and **(b)** the weighted averaging partial least squares (WA-PLS) method. Grey shaded histograms indicate the amount of variance of the fossil data explained by the 999 random environmental variables, whereas the black vertical lines indicate the variance explained by the reconstruction of summer productivity (VGPM) and summer SST with the respective method. The red dashed line represents the 95th percentile of the null distribution, the black dashed line the variance explained by the first RDA axis. Thus, the proportion of variance explained by the IK method for the reconstruction of summer SST from the fossil dataset is higher than 95 % ($p = 0.015$) of reconstructions of 999 random environmental variables. The summer SST reconstruction from the WA-PLS method is higher than 99 % of the random reconstructions ($p = 0.007$). The VGPM-based reconstruction of summer surface primary productivity is not significant (IK: $p = 0.76$; WA-PLS: $p = 0.16$).





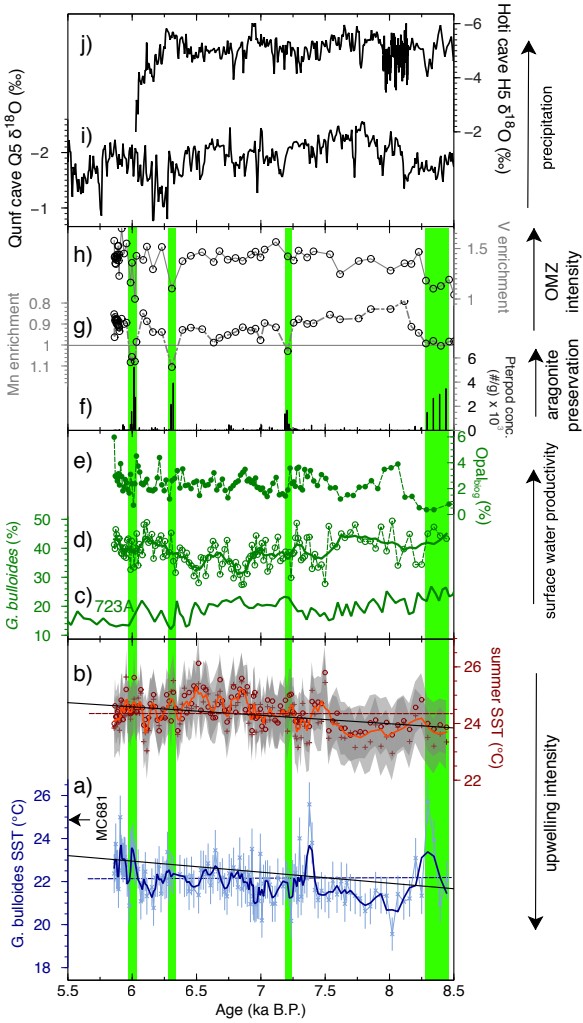

**Figure 6.** Time series of reconstructed SST from **(a)** the Mg/Ca ratio of *G. bulloides* (blue) indicating upwelling SST and **(b)** the consensus from the assemblage-based transfer function methods, IK and WA-PLS (red) indicating average summer SST (July–September). Thin lines represent real data, thick dashed lines the 3-pt running averages. Relative abundance of *G. bulloides* from ODP site 723A from Gupta et al. (2005) **(c)** and SL163 **(d)**, together with biogenic opal content **(e)**, indicates phases of increased surface water productivity. Aragonite

5  preservation is indicated by the concentration of pteropod fragments **(f)**, bottom-water oxygen conditions and state of the oxygen minimum zone (OMZ) is shown with relative enrichment or depletion of manganese **(g)** and vanadium **(h)**. Note the inverse relationship of both elements, as explained in the text. Stable oxygen isotopes of **(i)** Qunf cave (Fleitmann et al., 2007) and **(j)** Hoti cave stalagmites (Neff et al., 2001) expressing precipitation in Oman. Vertical solid green bars indicate periods of low biogenic opal content, concomitant with decreased OMZ intensity and increased aragonite preservation. Error bars for (a) were calculated by propagating the errors introduced by the

10  Mg/Ca measurements and the Mg/Ca-temperature calibration (Gaussian error propagation, see Mohtadi et al. (2014)). Grey shading gives



the uncertainty estimates for the transfer functions in (b), based on the sample specific root mean square error of prediction (RMSEP) using bootstrapping cross-validation.




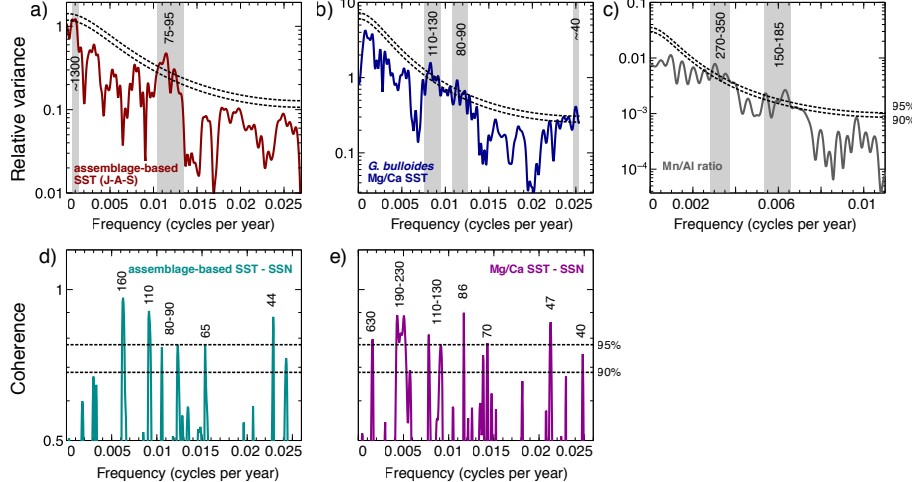

**Figure 7.** Results of the MTM spectral analysis show the frequency distribution and identified significant periodicities of the **(a)** assemblage-based SST record, **(b)** the SST record using Mg/Ca ratios of *G. bulloides* and **(c)** the Mn/Al ratio as a proxy for OMZ conditions. Black lines indicate 90 % and 95 % significance levels against red noise. Cross-spectral coherence of both SST records (**d** and **e**) with the record of sunspot numbers (Solanki et al., 2004) indicates that both records significantly covary on a wide range of frequencies. All time series were trend-removed prior to the analyses.





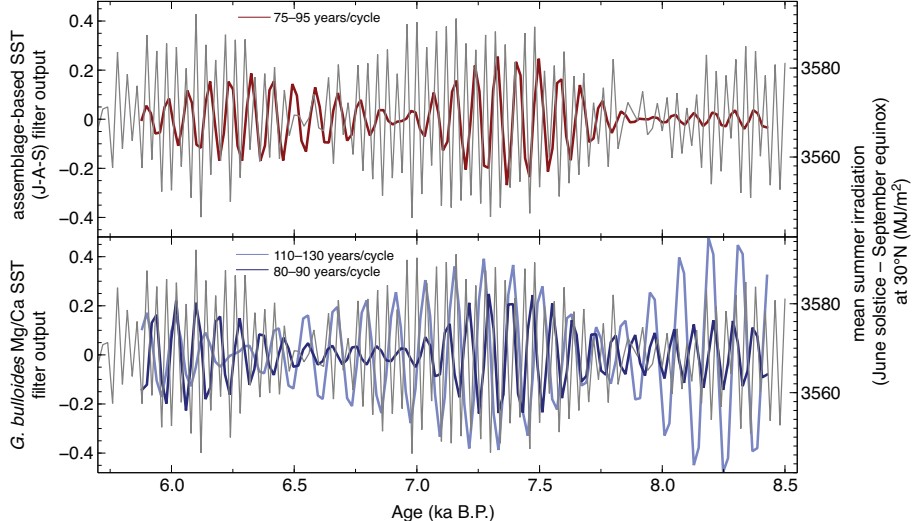

**Figure 8.** Comparison of filtered assemblage-based SST (red line, upper panel) and *G. bulloides* Mg/Ca SST (blue lines, lower panel) with the mean summer irradiance (grey line). Filtered components were extracted using a Gaussian band-pass filter with a central frequency of 0.0118 cycles year$^{-1}$ (~85 years cycle) and 0.0083 cycles year$^{-1}$ (~120-years cycle), respectively, and a bandwidth of 0.001 cycles year$^{-1}$. Solar irradiance was computed using the Laskar 2004 solution and a solar constant of 1365 W m$^{-2}$ with the program AnalySeries 2.0.8 (Paillard et al., 1996).





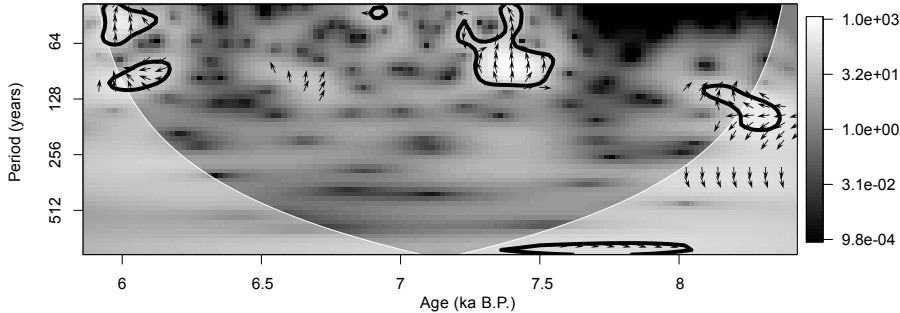

**Figure 9.** Cross wavelet transform (XWT) of the trend-removed Mg/Ca-based and assemblage-based SST time series. The 5 % significance level is shown as thick contours. Relative phasing of both records is indicated by arrows. Arrows within the significant intervals pointing mostly upwards, which indicates a consistent phase-shift of both records by 90° or 270°, respectively, of assemblage-based SST relative to Mg/Ca-based SST. The cone of influence, where edge artefacts might be introduced, is indicated by white shading.



**Table 1.** Samples for radiocarbon dating with the organic compound used, dating results and calibrated ages using the MARINE13 calibration curve (Reimer et al., 2013) and a reservoir correction of $\Delta R = 231 \pm 31$ years. Analyses were conducted at Beta Analytics in Miami/FL, U.S.A. (Beta) and the Leibniz Laboratory in Kiel, Germany (KIA).

| Lab Code | Core | Sample Type | Depth (cm) | Conventional $^{14}$C age (years) | Error | $\delta^{13}$C (‰) | cal. min (95%) | cal. max (95%) |
|---|---|---|---|---|---|---|---|---|
| Beta342813 | MC681 | mixed PF | 1 | 650 | 30 | –2.4 | | |
| Beta342812 | MC681 | bulk org. | 1 | 630 | 30 | –20.4 | | |
| Beta346603 | SL163 | bulk org. | 52.5 | 2100 | 30 | –19.4 | 1329 | 1505 |
| Beta346604 | SL163 | bulk org. | 58.5 | 5740 | 30 | –19.3 | 5788 | 5973 |
| KIA47119 | SL163 | *N. dutertrei* | 77.75 | 5760 | 30 | +0.24 | 5841 | 5987 |
| Beta319751 | SL163 | *N. dutertrei* | 141.25 | 5990 | 30 | –0.5 | 6095 | 6266 |
| Beta319752 | SL163 | *N. dutertrei* | 193.75 | 6350 | 40 | +0.5 | 6436 | 6646 |
| KIA47120 | SL163 | *N. dutertrei* | 252.75 | 6715 | 35 | +0.48 | 6878 | 7114 |
| Beta319753 | SL163 | *N. dutertrei* | 278.75 | 6670 | 40 | +0.8 | 6790 | 7043 |
| Beta319754 | SL163 | *N. dutertrei* | 295.0 | 7030 | 40 | +0.3 | 7246 | 7407 |
| Beta319755 | SL163 | *N. dutertrei* | 326.25 | 6990 | 40 | +0.4 | 7193 | 7386 |
| KIA47121 | SL163 | *N. dutertrei* | 353.75 | 7420 | 40 | +1.59 | 7568 | 7737 |
| KIA47122 | SL163 | *N. dutertrei* | 392.5 | 8090 | 40 | +1.42 | 8211 | 8396 |
| Beta319756 | SL163 | *N. dutertrei* | 395.0 | 8090 | 40 | +0.8 | 8211 | 8396 |
| Beta342816 | SL163 | *N. dutertrei* | 417.5 | 8500 | 40 | –0.4 | 8647 | 8957 |

mixed PF=mixed planktic foraminifera; bulk org.=bulk organic fraction



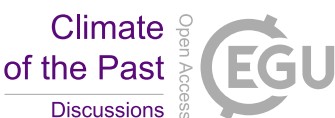

**Table 2.** Results of the redundancy analysis of the adjusted calibration dataset. Highest correlation with RDA1 axis and highest explanation of the PF assemblage variance is shown by temperature (bold numbers).

| | RDA1 | RDA2 | RDA3 | | |
|---|---|---|---|---|---|
| Eigenvalue | 1.763 | 0.185 | 0.097 | | |
| Cumulative proportion | | | | Captured variance | |
| of variance explained (%) | 35.9 | 39.6 | 41.6 | of species data | **Proportion (%)** |
| **Correlation** | | | | | |
| Temperature | **–0.93** | –0.07 | 0.15 | 1.55 | **31.5** |
| Salinity | –0.87 | 0.47 | –0.05 | 0.24 | 5.0 |
| Eppley-VGPM | 0.33 | 0.75 | 0.26 | 0.19 | 4.0 |
| uCbPM | –0.02 | –0.65 | 0.32 | 0.07 | 1.5 |
| Chlorophyll $\alpha$ | 0.39 | 0.36 | –0.18 | 0.05 | 1.0 |

VGPM=Vertical Generalized Production Models; uCbPM=updated Carbon-based Production Model.



**Table 3.** Cross-validated root mean squared error of prediction (RMSEP) as absolute values and relative to the range of the target variable, as well as the coefficient of determination. The results of the significance analysis after Telford and Birks (2011) from both transfer function methods are given, $p$ indicates the statistical significance against the null distribution of 999 random reconstructions.

| | RMSEP | RMSEP (% of target range) | $R^2$ | SST $p =$ | Eppley-VGPM $p =$ |
|---|---|---|---|---|---|
| Imbrie-Kipp | 0.92 | 15.72 | 0.67 | 0.015 | 0.758 |
| WA-PLS | 0.95 | 16.20 | 0.65 | 0.007 | 0.160 |

WA-PLS=weighted averaging partial least squares method.