# Peer review of "Decadal resolution record of Oman upwelling indicates solar forcing of the Indian summer monsoon (9-6 ka)"

_Climate of the Past, 2016_

## Referee Comment (RC1) · Anonymous Referee #1 · 13 Dec 2016

The manuscript by Munz and others " Decadal resolution record of Oman margin upwelling indicates persistent solar forcing of the Indian summer monsoon after the early Holocene summer insolation maximum" is based on sediment cores from the Oxygen Minimum Zone (OMZ) and upwelling cell from the Oman margin and covers about 3.000 years of climatic and paleoceanographic history during the Holocene. The manuscript is well written and easy to read. The reconstruction is based on several methods such as faunal assemblages, foraminiferal Mg/Ca ratios and bulk sediment geochemistry. Bases on these records the authors identify the presence of a solar activity cycle and therefore suggest atmospheric forcing controls OMZ dynamics. It is a

nice study and will make a step forward in understanding this complex region. However, the discussion lacks some thoughts about the age model and significance of the data as well as comparisons to other published datasets. Nevertheless, the study certainly fits into the scope of CP and should be considered for publication after minor/moderate revisions.

Major comment:

1) There are two age reversals within the sediment core. It is okay, if you can fit a smooth spline model to it which results into continuous depositions rates. However, to my opinion, considering the sample resolution of about 19 years and the observation of the Gleissberg cycle this needs some more discussions. 2) Many studies in the past recent years have demonstrated the impact of solar forcing on paleoceanographic and climatic records (e.g Moffa-Sanchez et al. 2014; Knudsen et al. 2011). Total solar irradiance (TSI) is controlled by different cycles such as the shorter Gleissberg (87 ys) and the longer de Vries (210 ys) cycle. The latter is not dominant in the present records. I wonder why the spectral analyses reveals the shorter Gleissberg and not also the de Vries cycle as this was clearly shown by other studies (e.g Steinhilber et al., 2012; Moffa-Sanchez et al. 2014 etc). This may give a hint that this is a statistical artefact as discussed by Turner et al. (2015), especially for cycles ranging between 120-140 years. 3) The authors conclude that atmospheric forcing (solar forcing) is the origin for OMZ dynamics rather than intermediate water mass dynamics. Also modelling results reveal a response of intermediate water masses to solar forcing (Seidenglanz et al. 2012). However, to state something like this the authors should compare their record to other paleoceanographic records (if available at this resolution) and not only to stalagmite records.

Other comments on the manuscript: Page 2 Line 18: There are studies revealing SST variations probably forced by changes in total solar radiation in the North Atlantic (Moffa-Sanchez et al. 2014).

Page 3 Line 10: What are the oxygen concentrations? Line 14: What are the salinities?

Page 4 Line 10: It is not appropriate to cite only the website you should refer here to the original study

Page 4 Line 7-13: I am a bit worried about the error of the $\Delta$R as the authors claim to see the Gleissberg cycle of about 87 years, which is nearly the same compared to the overall error of the $\Delta$R.

Page 5 Line 24: I think the ECRM 752-1 should read 3.761 (Greaves et al. 2008). As the authors discuss Mg/Ca based SST variability of less than 2°C could the authors provide an error for the temperature reconstruction? Line 32: What do the correlations say between the individual elemental/Ca ratios against Mg/Ca? What about Al/Ca and Fe/Mg ratios? As the authors discuss later Mn/Al ratios from bulk analyses it would be nice to know the variability of the Al/Ca ratios.

Page 6 Line 2: A fragmentation index not only tells us something about dissolution, but also about changing bottom water current strength. If there are strong currents at 600m water depth these might transport lighter particles, which in turn would indicate less dissolution. I do not believe this study has to tackle severe dissolution problems, but I think a fragmentation index is not an appropriate proxy for that.

Page 7 Line 10: Instability or strong bottom currents? Similar as off the Peruvian margin (Erdem et al 2016)?

Line 5: How do the pteropods look like? I think a better indication is the in situ carbonate saturation. There is data available to calculate the in situ $\Delta CO_3^{2-}$! Interestingly, there is a sharp increase in alkalinity at 1000m water depth, can this be explained by the dissolution of the pteropods? (Jansen et al. 2002).

Page 10 Line 4: Can this statement be proven by a comparison to the cosmogenic nuclides record of Steinhilber et al. (2012)? Moreover, modelling results for the North Atlantic suggest that the phase shift of TSI on SST is about 40 years (Seidenglanz et

al. 2012). What would you expect for your location? Can the authors comment on that?

Line 10: The authors should cite here earlier studies that made similar observations.

Page 11: Line 3: Mn has been introduced, but what about Al? How does the Mn/Al downcore record look like? What is the variability of Al? Can the authors please clarify.

Figures and tables

Figure 9: This does not really look convincing to me. Maybe the authors should consider to show also a simple evolutionary spectra revealing the intensities of the present cyclicity through time.

Table 1.

The error should read $\pm$

---

## Short Comment (SC1) · 18 Dec 2016

This is an excellent contribution which demonstrates once again the great importance of solar activity changes as climate drivers in this region. The new high-resolution dataset adds another very useful case study.

It would be good if the authors can add a paragraph on the longer-term Holocene context of their time series. The described natural variability and cyclicity forms part of so-called millennial-scale cycles which have been first described in more detail by Bond et al. 2001 in their pionier paper in Science. Meanwhile, similar Holocene milennial-scale

cycles have been reported from many other places around the world, including the Arabian Sea. See a literature overview in our recent paper on this subject, pages 289-299: https://www.researchgate.net/publication/308928345_The_Sun%27s_Role_in_Climate

Interestingly, most of these studies link the millennial-scale cycles to solar activity changes. It would be good to see a comparison of your 2500 year long time series with the millennial-scale cycles of Bond et al. 2001 which form an important reference for Holocene climate evolution.

In this context, you might also consider a comparison with Menzel et al. 2014 who documented millennial-scale climate cycles with repeated dry/wet shifts from a central Indian lake. http://www.sciencedirect.com/science/article/pii/S0031018214003009

---

## Referee Comment (RC2) · Anonymous Referee #2 · 19 Dec 2016

Review of manuscript "Decadal-resolution record of Oman margin upwelling indicates persistent solar forcing..." by Munz et al.. This manuscript presents data from a sediment core from the northern Arabian Sea. The high-resolution data spanning the early to mid Holocene comprise abundance records of planktic foraminifera and SST estimates based on those, Mg/Ca based SST estimates, some element concentration records and frequency analyses of a number of downcore profiles. Overall this is an interesting paper touching on an important part of ongoing research. The manuscript is easy to read and the quality of the figures is high. With regard to the main thrust of this work, relating changes in the monsoon system to variations in solar insolation

at the decadal to century scale would indeed be relevant in the context of understanding the processes having driven past climate change. It would also aid in improving predictions of future climate change. There are a number of concerns that prevent recommending publication in the current form. The first issue relates to the age control of core SL 163. The authors correctly report that there are age reversals in the section of the core presented in this study. It is to some degree justified to apply a spline to the AMS14C dates and assume constant sedimentation occurred. It seems, though, that the location is not free of the potential for disruption of normal sedimentation. The authors themselves mention that a section of the top of the core is missing (section 4.1) alluding to slope instability as a possible cause. If so, is it possible that the sections with age reversals represent other periods of sediment instability (turbidites)? This should be discussed in more detail.

The authors state that the overall time resolution across the entire section presented in this manuscript is around 20 years. Whilst undoubtedly true for the section younger than 7.5 KaBP, it is not true for the section covering the time period between 8.5 and 7.5 KaBP. The sample resolution in this section seems much closer to 50 years. This should be clarified. Statements related to highlighting differences in the spectral analyses results for different time periods should be double checked as well (see for example page 11 lines 16-18). To me, the fact that higher frequencies have not been recorded between 8.5 and 7.5 KaBP is likely a result of the insufficient time resolution in this section.

There is a little bit of confusion related to a statement made on figure 8 (page 10 lines 4-6). Based on the text a rather strict relation between solar insolation and SST change has been found, i.e. solar insolation leading SST change by roughly 200 years. I may read figure 8 incorrectly, but does this figure not partly show the opposite relation. Between 7.7 and 6.7 KaBP for example my read of the figure implies that amplitude variation in SST (certainly for the assemblage based data) leads the respective change in solar insolation. Other sections of the record also do not show the alleged relation.

This part of the manuscript should be revised. This should also have ramifications with regard to the relation of solar forcing and the response in the climate system. The authors spend quite a bit of text on the technical details related to SST analysis. It would be useful to know what the error bar is with regard to both SST estimates (I might have overlooked a statement on this). My best guess is that it is in the ballpark of $\pm$1-2 degC. Using such an uncertainty, quite a bit of the SST variability would not represent statistically significant change. With regard to the Mg/Ca based SST estimates the main thrust of this paper might change to work assessing specific short term events with significant changes in temperature (e.g. around 8.2 KaBP). Please note that the statement on page 9 line 8 is misleading. There is a maximum change in temperatures of roughly 6 degC, the majority of the change is, however, much smaller. Overall this is an interesting paper, that merits publication after a moderate revision. Minor issues:

Title is too long: (suggestion) Solar forced decadal-scale upwelling in the Arabian Sea during the early Holocene. Abstract: The first four lines are misleading. The abstract should introduce solar driven climate change and not the socio-political implications thereof.

Introduction: There are some detailed statements related to monsoon circulation. in the Arabian Sea (page 2 lines 2-8) that would be better placed at the start of chapter 2. In the introduction more generic wording could be used.

Importance of AAIW in the Arabian Sea (page 3 lines 11-18). Whilst Boening and Bard indeed suggest that there is a strong influence of AAIW in the Arabian Sea, most of the available work seems to suggest that AAIW in the modern ocean does not reach the Arabian Sea (at least not with near pristine properties).

Figure 1: labeling partly too small.

Figure 6: Labeling on the right side should have the same orientation.

---

## Author Comment (AC1) · 10 Mar 2017

We would like to thank the two anonymous referees and Sebastian Luening for their constructive comments, their time and effort which helped to improve the quality of the manuscript. The main issues, as expressed by both anonymous referees, is concerning the age model and significance of the data presented in our study. We are aware of these points and added a more critical discussion about the age model and significance of our data in the revised version of the manuscript. Below is a point-by-point statement of the reviewer comments and our responses supplemented with a revised

version of our manuscript indicative of the changes made to the text.

Authors response to referee #1 1) There are two age reversals within the sediment core. It is okay, if you can fit a smooth spline model to it which results into continuous depositions rates. However, to my opinion, considering the sample resolution of about 19 years and the observation of the Gleissberg cycle this needs some more discussions.

> We thank the reviewer for this comment and added a more critical discussion of potential artefacts from the age model. The offset of the reversals is relatively low (<70 and <40 years) and we therefore hypothesize that either changing upwelling intensities or changes in intermediate water mass, as it was shown from the Peruvian upwelling (Fontugne et al., 2004), or bioturbation processes ages could be responsible for the slight offset of the radiocarbon dates (see p.8, l. 3-8). However, as the two age reversals are no clear outliers from the smooth spline fit and we did not find evidence for interruptions of the sedimentary succession (see response to comment below), we are confident that our age model represents a continuous deposition. Further, we are aware of the overall error of the reservoir correction factor of $\pm31$ years being close to the observed frequencies (75-95 years and 80-90 years) of the supposed Gleissberg cycle. However, we also found a 110-130 year cyclicity in the Mg/Ca-SST record, which is close to another prominent sunspot cycle of $\sim$132 years, probably a subharmonics of the Hale cycle (Attolini et al. 1990). We added this to the discussion on p. 11.

2) Many studies in the past recent years have demonstrated the impact of solar forcing on paleoceanographic and climatic records (e.g Moffa-Sanchez et al. 2014; Knudsen et al. 2011). Total solar irradiance (TSI) is controlled by different cycles such as the shorter Gleissberg (87 ys) and the longer de Vries (210 ys) cycle. The latter is not dominant in the present records. I wonder why the spectral analyses reveals the shorter Gleissberg and not also the de Vries cycle as this was clearly shown by other studies (e.g Steinhilber et al., 2012; Moffa-Sanchez et al. 2014 etc). This may give a hint that this is a statistical artefact as discussed by Turner et al. (2015), especially for cycles

ranging between 120-140 years.

> The apparent absence of a ∼200-year cyclicity is a common finding also known from other studies of Asian monsoon records. Studies using stable isotopes off Japan (Sagawa et al., 2014) and in the South China Sea (Wang et al., 1999), found cycles in their data matching cycles of the TSI record, except the 210-year De Vries cycle, although being very prominent in the TSI record. Duan et al. (2014) showed with a speleothem record from Dongge cave (China), that coherency with the Gleissberg cycle is persistent over the last 4 ka, whereas the De Vries cycle occurred only over a short 1 ka long period. We therefore hypothesize that the De Vries cycle is not detectable in our record, due to a transient link of the solar-monsoon relationship, potentially suppressed by changes in the ocean-atmosphere system in the ENSO domain (Berkelhammer et al., 2010). We added this to the discussion on p. 11, l. 5-14. However, our Mg/Ca-based SST record shows strongest coherency with the SSN record of Solanki et al. (2004) on both, the ∼88-year Gleissberg and the ∼190-230 year De Vries frequency, as shown in Fig. 7e. See p. 11, l. 18

3) The authors conclude that atmospheric forcing (solar forcing) is the origin for OMZ dynamics rather than intermediate water mass dynamics. Also modelling results reveal a response of intermediate water masses to solar forcing (Seidenglanz et al. 2012). However, to state something like this the authors should compare their record to other paleoceanographic records (if available at this resolution) and not only to stalagmite records.

> We thank the reviewer for the constructive suggestion. We added a comparison with other high-resolution records from the Arabian Sea (Staubwasser et al., 2003 and Deplazes et al., 2013) to the revised version of the manuscript (see Fig. 7 and p. 10, l. 18)

Other comments on the manuscript: Page 2 Line 18: There are studies revealing SST variations probably forced by changes in total solar radiation in the North Atlantic

(Moffa-Sanchez et al. 2014).

> We thank the reviewer for pointing out this ambiguity. Of course we refer here to SST records from the Arabian Sea, which are, to our knowledge, not available at a decadal-scale resolution. We clarified this in the text.

Page 3 Line 10: What are the oxygen concentrations? Line 14: What are the salinities?

> We added oxygen (0.07-0.52 ml/l) and salinity (35.5-36.8 psu) values of the intermediate waters from the literature (Emery and Meincke, 1986; You, 1997) on p. 3, l. 19 and 21.

Page 4 Line 10: It is not appropriate to cite only the website you should refer here to the original study.

> We thank the reviewer for this comment and refer now directly to Southon et al. (2002) and von Rad et al. (1999).

Page 4 Line 7-13: I am a bit worried about the error of the deltaR as the authors claim to see the Gleissberg cycle of about 87 years, which is nearly the same compared to the overall error of the deltaR.

> Please see the response to first major comment.

Page 5 Line 24: I think the ECRM 752-1 should read 3.761 (Greaves et al. 2008). As the authors discuss Mg/Ca based SST variability of less than 2 degC could the authors provide an error for the temperature reconstruction?

> The ECRM value of 3.75 is correct, 3.761 was a value after removal of one result with a lower value, but without any indication that it was a wrong measurement (Table 4 in Greaves et al. 2008). We also added the error estimation using error propagation (Mohtadi et al., 2014) to the methods section, which was previously mentioned in caption of Fig. 6. The overall average of the propagating 1-sigma error is 0.89 degC.

Line 32: What do the correlations say between the individual elemental/Ca ratios

against Mg/Ca? What about Al/Ca and Fe/Mg ratios? As the authors discuss later Mn/Al ratios from bulk analyses it would be nice to know the variability of the Al/Ca ratios.

> We added the correlations of other element/Ca vs. Mg/Ca values to page 6, l. 5. Al/Ca (r=0.06) and Fe/Ca (r=0.16) are not correlated to Mg/Ca, Mn/Ca has weak correlations (r=0.36).

Page 6 Line 2: A fragmentation index not only tells us something about dissolution, but also about changing bottom water current strength. If there are strong currents at 600m water depth these might transport lighter particles, which in turn would indicate less dissolution. I do not believe this study has to tackle severe dissolution problems, but I think a fragmentation index is not an appropriate proxy for that.

> We agree with the reviewer and assume that dissolution did not have any effect on the samples, which is also supported by the presence of pteropods. Undercurrents related to the upwelling process, however, are probably shallower (<200m). We therefore removed the fragmentation index from the text.

Page 7 Line 10: Instability or strong bottom currents? Similar as off the Peruvian margin (Erdem et al 2016)?

> We did not find any evidence for interruptions of the stratigraphical record in the form of slumps, turbidites, erosional surfaces or phosphorites, below the unconformity at 56 cm core depth. We therefore assume that the record we present here is homogeneous and undisturbed. The diatomaceous ooze above the unconformity at 56 cm core depth of SL163 showed an excursive increase of the sedimentary water content, relative to section below 56 cm, of more than 17%. We therefore assume this might have caused a decrease of gravitational stability of the sediment. However, we feel the assessment of a potential mechanism for the observed hiatus above the interval we are focusing on is beyond the scope of this study and will be presented elsewhere.

Line 5: How do the pteropods look like? I think a better indication is the in situ carbonate saturation. There is data available to calculate the in situ _CO32-! Interestingly, there is a sharp increase in alkalinity at 1000m water depth, can this be explained by the dissolution of the pteropods? (Jansen et al. 2002).

> The paper by Jansen et al. (2002) is dealing about the North Pacific. The authors state, that carbonate/aragonite saturation is much lower there compared to the Atlantic and Indian Ocean. Thus aragonite dissolution can occur in the water column, as the depth where the in-situ carbonate saturation becomes <1 is shallower. However, as pteropds are always present in our record, except for 11 samples where they were probably diluted due to high concentrations of foraminiferal shells, there was no aragonite dissolution in the water column, nor in the sediment. We therefore expect that foraminifera are not affected by dissolution, as they build calcite shells, the more dissolution resistant polymorph of calcium carbonate compared to aragonite.

Page 10 Line 4: Can this statement be proven by a comparison to the cosmogenic nuclides record of Steinhilber et al. (2012)? Moreover, modelling results for the North Atlantic suggest that the phase shift of TSI on SST is about 40 years (Seidenglanz et al. 2012). What would you expect for your location? Can the authors comment on that?

> The study by Seidenglanz et al. (2012) revealed a response of Arabian Sea surface temperatures to an idealized TSI forcing within less than 20 years for both idealized cycles, 90 and 200 years, which is probably not resolvable by our 19-year record.

Line 10: The authors should cite here earlier studies that made similar observations.

> We agree with the reviewer and cited previous studies using speleothem (Neff et al, 2001; Burns et al., 2002; Fleitmann et al., 2003; Dykoski et al., 2005; Wang et al., 2005; Duan et al. 2014) and marine records (von Rad et al., 1999; Wang et al., 1999; Gupta et al., 2005).

Page 11: Line 3: Mn has been introduced, but what about Al? How does the Mn/Al downcore record look like? What is the variability of Al? Can the authors please clarify.

> We used the enrichment factor relative to the detrital background ("average shale" as reported by Wedepohl, 1971, 1991) to decipher a possible secondary alteration of the geochemical composition. The enrichment factor is calculated as the element/Al ratio of the sample divided by the element/Al ratio of average shale. Hence, element/Al ratio and enrichment factor are covarying. We added this clarification to the methods section.

Figures and tables Figure 9: This does not really look convincing to me. Maybe the authors should consider to show also a simple evolutionary spectra revealing the intensities of the present cyclicity through time.

> We added a wavelet power spectrum of both SST time series to demonstrate the evolution of the dominant signals through time.

Table 1. The error should read $\pm$

> We added the correct sign to the column header.

Authors response to referee #2 The authors themselves mention that a section of the top of the core is missing (section 4.1) alluding to slope instability as a possible cause. If so, is it possible that the sections with age reversals represent other periods of sediment instability (turbidites)? This should be discussed in more detail.

> We thank the reviewer for this comment and discuss possible disturbances of the stratigraphical record in more detail. Please see the answer to referee #1 comments.

The authors state that the overall time resolution across the entire section presented in this manuscript is around 20 years. Whilst undoubtedly true for the section younger than 7.5 KaBP, it is not true for the section covering the time period between 8.5 and 7.5 KaBP. The sample resolution in this section seems much closer to 50 years. This should be clarified.

> We agree with the reviewer that the sample spacing is larger in the lower part of the record. The lowermost 15 cm of the core (six samples) span 287 years of deposition, in fact close to a spacing of 50 years. We added this for clarification to the text (p. 7, l. 22). However, the overall average sample spacing is 19.9 years for the Mg/Ca-based SST time series, 15.3 years in the upper half and 28.9 years in the lower half of the record. The assemblage-based SST record has a slightly higher average resolution of 18.4 years, as there were no samples removed due to potential contamination.

Statements related to highlighting differences in the spectral analyses results for different time periods should be double checked as well (see for example page 11 lines 16-18). To me, the fact that higher frequencies have not been recorded between 8.5 and 7.5 KaBP is likely a result of the insufficient time resolution in this section.

> We included a continuous wavelet transform of both SST time series, which shows significant periodicities of the Mg/Ca-SST record on the ∼80- to ∼130-year bandwidth within the lowermost part of the record (see Fig. 10).

There is a little bit of confusion related to a statement made on figure 8 (page 10 lines 4-6). Based on the text a rather strict relation between solar insolation and SST change has been found, i.e. solar insolation leading SST change by roughly 200 years. I may read figure 8 incorrectly, but does this figure not partly show the opposite relation. Between 7.7 and 6.7 KaBP for example my read of the figure implies that amplitude variation in SST (certainly for the assemblage based data) leads the respective change in solar insolation. Other sections of the record also do not show the alleged relation. This part of the manuscript should be revised. This should also have ramifications with regard to the relation of solar forcing and the response in the climate system.

> We agree with the reviewer that the statements made about Fig. 8 are misleading and that this section has to be revised. Our statements are mistakenly based on another version of the figure, which compares filter outputs of both SST time series with filter outputs of the reconstructed total solar irradiance based on cosmogenic nuclides

(Steinhilber et al., 2012). The version of Fig. 8 in the manuscript, shows solar energy based solely on orbital solutions. Our findings from spectral analysis and the following interpretation regarding solar forcing of the monsoon system is, however, focusing on the solar cycle and sunspot activity, which are expressed in the TSI record. We corrected and clarified this in the revised manuscript (p. 11, l. 26-33) and changed Fig. 8 accordingly (now numbered Fig. 9). Although there is a slight offset in some parts of the record, the new figure shows that the amplitude modulation of both SST time series filtered on the 88-year frequency shows a maximum centered at ∼7.4 ka B.P., which is co-varying with a modulation maximum of the 88-year band-pass filtered time series of TSI. On the 132-year bandwidth, however, the Mg/Ca-based SST time series and band-pass filtered TSI are offset by ∼400 years. Modelling results indicate a response of surface temperatures to solar forcing within less than 50 years (Seidenglanz et al., 2012). We therefore assume no direct forcing of Arabian Sea upwelling intensity from the ∼132-year solar cycle.

The authors spend quite a bit of text on the technical details related to SST analysis. It would be useful to know what the error bar is with regard to both SST estimates (I might have overlooked a statement on this). My best guess is that it is in the ballpark of 1-2 degC. Using such an uncertainty, quite a bit of the SST variability would not represent statistically significant change. With regard to the Mg/Ca based SST estimates the main thrust of this paper might change to work assessing specific short term events with significant changes in temperature (e.g. around 8.2 KaBP).

> Error bars for both SST time series are given in Fig. 6a/b. Error assessment for the Mg/Ca-based SST is based on error propagation (Mohtadi et al., 2014). It yields an average propagating 1-sigma error of 0.89 degC. We clarified this and added error estimation methods to the text (p. 6, l. 9 and p. 9, l. 5; previously only found in the caption of Fig. 6). Error estimates for the assemblage-based SST reconstructions are given in Table 3 (RMSEP=0.92-0.95). We expanded the discussion towards the point whether the most noticeable features of the Mg/Ca-SST record centered at ∼5.9, ∼7.4
and ∼8.2 ka B.P. could be related to North Atlantic ice rafting events, i.e., Bond events 4 and 5 (Bond et al., 2001), which were shown to affect Arabian Sea upwelling intensity (Gupta et al., 2003). See discussion on p. 10, l. 24-31.

Please note that the statement on page 9 line 8 is misleading. There is a maximum change in temperatures of roughly 6 degC, the majority of the change is, however, much smaller.

> We clarified this in the text and state that the range of the Mg/Ca-SST is 6.16 degC and the mean absolute deviation is 0.72 degC (p. 10, l. 7).

Minor issues: Title is too long: (suggestion) Solar forced decadal-scale upwelling in the Arabian Sea during the early Holocene.

> We agree with the reviewer and shortened the title to "Decadal resolution record of Oman upwelling indicates solar forcing of the Indian summer monsoon (9-6 ka)"

Abstract: The first four lines are misleading. The abstract should introduce solar driven climate change and not the socio-political implications thereof.

> We moved this to the Introduction (p. 1, l. 17)

Introduction: There are some detailed statements related to monsoon circulation. in the Arabian Sea (page 2 lines 2-8) that would be better placed at the start of chapter 2. In the introduction more generic wording could be used.

> These statements introducing the general development of the monsoonal winds are leading to the establishment of the oxygen minimum zone. We agree with the reviewer that some of these statements are specific and could be placed in Chap. 2, but still think the upwelling process has to be introduced prior to introducing OMZ.

Importance of AAIW in the Arabian Sea (page 3 lines 11-18). Whilst Boening and Bard indeed suggest that there is a strong influence of AAIW in the Arabian Sea, most of the available work seems to suggest that AAIW in the modern ocean does not reach the

Arabian Sea (at least not with near pristine properties).

> We thank the reviewer for pointing this out and clarified it in the text (p. 3, l. 17).

Figure 1: labeling partly too small.

> Labeling of Fig. 1, which was larger in a previous version, was increased. Thank you.

Figure 6: Labeling on the right side should have the same orientation. > We thank the reviewer for indicating this and changed the orientation of the labeling.

Authors response to short comment #1 It would be good if the authors can add a paragraph on the longer-term Holocene context of their time series. The described natural variability and cyclicity forms part of socalled millennial-scale cycles which have been first described in more detail by Bond et al. 2001 in their pionier paper in Science. Meanwhile, similar Holocene milennial-scale cycles have been reported from many other places around the world, including the Arabian Sea. See a literature overview in our recent paper on this subject, pages 289-299: https://www.researchgate.net/publication/308928345_The_Sun%27s_Role_in_Climate Interestingly, most of these studies link the millennial-scale cycles to solar activity changes. It would be good to see a comparison of your 2500 year long time series with the millennial-scale cycles of Bond et al. 2001 which form an important reference for Holocene climate evolution. In this context, you might also consider a comparison with Menzel et al. 2014 who documented millennial-scale climate cycles with repeated dry/wet shifts from a central Indian lake. http://www.sciencedirect.com/science/article/pii/S0031018214003009

> We thank Sebastian Luening for his suggestion and added a comparison with the North Atlantic drift ice record to the manuscript. In fact, the most noticeable changes of the Mg/Ca-SST record of G. bulloides occur at ∼5.9, ∼7.4 and ∼8.2 ka B.P., which are partly in concordance with the North Atlantic Bond events 4 (5.9 ka) and 5 (8.2 ka),

as well a less pronounced event at 7.4 ka. However, to significantly assess millennial-scale cycles in our record, the time span covered of ~2.5 k years is likely too short (see discussion p. 10, l. 24-31).

Additional References Attolini, M.R., Cecchini, S., Galli, M. and Nanni, T. On the persistence of the 22 y solar cycle, Solar Physics, 125(2), 389, doi:10.1007/bf00158414, 1990.

Berkelhammer, M., Sinha, A., Mudelsee, M., Cheng, H., Edwards, R.L. and Cannariato, K. Persistent multidecadal power of the Indian Summer Monsoon, Earth and Planetary Science Letters, 290(1-2), 166, doi:10.1016/j.epsl.2009.12.017, 2010.

Burns, S. J. et al.: A 780-year annually resolved record of Indian Ocean monsoon precipitation from a speleothem from south Oman, Journal of Geophysical Research, 107, 2002.

Deplazes, G., Lückge, A., Peterson, L. C., Timmermann, A., Hamann, Y., Hughen, K. A., Röhl, U., Laj, C., Cane, M. A., Sigman, D. M., and Haug, G. H.: Links between tropical rainfall and North Atlantic climate during the last glacial period, 6, 1–5, 2013.

Duan, F., Wang, Y., Shen, C.-C., Wang, Y., Cheng, H., Wu, C.-C., Hu, H.-M., Kong, X., Liu, D and Zhao, K. Evidence for solar cycles in a late Holocene speleothem record from Dongge Cave, China., Scientific Reports, 4, 5159, doi:10.1038/srep05159, 2014.

Dykoski, C. A., Edwards, R. L., Cheng, H., Yuan, D., Cai, Y., Zhang, M., Lin, Y., Qing, J., An, Z., and Revenaugh, J.: A high-resolution, absolute-dated Holocene and deglacial Asian monsoon record from Dongge Cave, China, Earth and Planetary Science Letters, 233, 71–86, 2005.

Emery, W.J. and Meincke, J. Global Water Masses - Summary and Review, Oceanologica Acta, 9(4), 383-391, 1986.

Fleitmann, D., Burns, S. 5 J., Mudelsee, M., Neff, U., Kramers, J., Mangini, A., and Matter, A.: Holocene forcing of the Indian monsoon recorded in a stalagmite from

Southern Oman, Science, 300, 1737–1739, 2003.

Fontugne, M., Carre, M., Bentaleb, I., Julien, M. and Lavallee, D. Radiocarbon reservoir age variations in the south Peruvian upwelling during the Holocene., Radiocarbon, 46(2), 531-537, 2004.

Gupta, A. K., Anderson, D. M., and Overpeck, J. T.: Abrupt changes in the Asian southwest monsoon during the Holocene and their links to the North Atlantic Ocean, Nature, 421, 354–357, 2003. Gupta, A. K., Das, M., and Anderson, D. M.: Solar influence on the Indian summer monsoon during the Holocene, Geophysical Research Letters, 32, L17 703, 2005.

Mohtadi, M., Prange, M., Oppo, D.W., De Pol-Holz, R., Merkel, U., Zhang, X., Steinke, S. and Lückge, A. North Atlantic forcing of tropical Indian Ocean climate, Nature, 509(7498), 76-80, doi:10.1038/nature13196, 2014.

Neff, U., Burns, S. J., Mangini, A., Mudelsee, M., Fleitmann, D., and Matter, A.: Strong coherence between solar variability and the monsoon in Oman between 9 and 6 kyr ago., Nature, 411, 290–293, 2001.

Sagawa, T., Kuwae, M., Tsuruoka, K., Nakamura, Y., Ikehara, M. and Murayama, M. Solar forcing of centennial-scale East Asian winter monsoon variability in the mid- to late Holocene, Earth and Planetary Science Letters, 395, 124-135, doi:10.1016/j.epsl.2014.03.043, 2014.

Solanki, S.K., Usoskin, I.G., Kroomer, B., Schüssler, M. and Beer, J. Unusual activity of the Sun during recent decades compared to the previous 11,000 years., Nature, 431(7012), 1084-1087, doi:10.1038/nature02995, 2004.

Southon, J., Kashgarian,M., Fontugne,M.,Metivier, B., and Yim,W.W. S.:Marine Reservoir Corrections for the Indian Ocean and Southeast Asia, Radiocarbon, 44, 167, 2002.

Staubwasser, M., Sirocko, F., Grootes, P. M., and Segl, M.: Climate change at the 4.2 ka BP termination of the Indus valley civilization and Holocene south Asian monsoon

variability, Geophysical Research Letters, 30, 2003.

von Rad, U., Schaaf, M., Michels, K.H., Schulz, H., Berger W.H. and Sirocko, F. A 5000-yr Record of Climate Change in Varved Sediments from the Oxygen Minimum Zone off Pakistan, Northeastern Arabian Sea, Quaternary Research, 51(01), 39, doi:10.1006/qres.1998.2016, 1999.

Wang, L., Sarnthein, M., Erlenkeuser, H., Grimalt, J., Grootes, P., Heilig, S., Ivanova, E., Kienast, M., Pelejero, C., and Pflaumann, U.: East Asian monsoon climate during the Late Pleistocene: high-resolution sediment records from the South China Sea, Mar. Geol., 156, 245, 1999.

Wang, Y., Cheng, H., Edwards, R. L., He, Y., Kong, X., and An, Z.: The Holocene Asian monsoon: links to solar changes and North Atlantic climate, Science, 308, 854–857, 2005.

Wedepohl, K. H.: Environmental influences on the chemical composition of shales and clays, in: Physics and Chemistry of the Earth, edited by Ahrens, L. H., Press, F., Runcorn, S. K., and Urey, H. C., pp. 307–331, Oxford, 1971.

Wedepohl, K. H., Merian, E., Anke, M., Ihnat, M., and Stoeppler, M.: The Composition of Earth's Upper Crust, Natural Cycles of Elements, Natural Resources, Wiley-VCH Verlag GmbH, Weinheim, 1991. Southon, J., Fontugne, M., Kashgarian, M., Metivier, B. and Yim, W.W.S. Marine reservoir corrections for the Indian Ocean and Southeast Asia, 44(1), 167-180, 2002.

Steinhilber, F., Abreu, J. A., Beer, J., Brunner, I., Christl, M., Fischer, H., Heikkilä, U., Kubik, P.W., Mann, M., McCracken, K. G., Miller, H., Miyahara, H., Oerter, H., and Wilhelms, F.: 9,400 years of cosmic radiation and solar activity from ice cores and tree rings., Proceedings of the National Academy of Sciences of the United States of America, 109, 5967–5971, 2012.

You, Y. Seasonal variations of thermocline circulation and ventilation in the Indian Ocean, Journal of Geophysical Research: Oceans, 102(C5), 10391, doi:10.1029/96jc03600, 1997.

Please also note the supplement to this comment:
http://www.clim-past-discuss.net/cp-2016-107/cp-2016-107-AC1-supplement.pdf

**Supplement:**

**Decadal resolution record of Oman  upwelling indicates  solar forcing of the Indian summer monsoon (9-6 ka)**

[revised manuscript text omitted]
. No correlations exist between Mg/Ca and element/Al ratios of Al/Ca ($r = 0.06$) and Fe/Ca ($r = 0.16$) and only weak correlations for Mn/Ca ($r = 0.36$). Paleotemperature estimates based on trace elemental concentrations in foraminiferal calcite can potentially suffer from post-depositional dissolution and pref-

15 erential removal of more solution susceptible Mg-rich calcite (e.g., Brown and Elderfield, 1996; Dekens et al., 2002). ~~We therefore tested a systematic dissolution bias of samples from core SL163 using a cross correlation of Mg/Ca ratios and the fragmentation index of Le and Shackleton (1992). If Mg/Ca measurements were affected by carbonate dissolution, a strong negative relationship between Mg/Ca ratios and fragmentation indices would be expected, which is not the case for our samples ($r = -0.07$, $p = 0.43$). Furthermore, the presence of pteropods~~ Pteropod concentration was always >0, except for 11 samples

20 where they were likely diluted by a high concentration of foraminiferal shells. The persistent presence of aragonite indicates that selective dissolution  could not affect Mg/Ca-ratios, as planktic foraminifera build their shells from calcite, the more dissolution resistant polymorph of calcium carbonate compared to aragonite.

**3.4 Bulk geochemical analyses**

Bulk geochemical analyses of the sediment were performed on average every 5 cm with X-ray fluorescence (XRF). Concen-

25 trations of manganese (Mn) and vanadium (V) were quantitatively analysed as an indicator for bottom water redox conditions and state of the OMZ. After fusion of the samples with lithium metaborate at $1200\,^\circ$C for 20 minutes (sample/LiBO$_2$ = 1/5) samples were measured using Philips PW 2400 and PW 1480 wavelength dispersive spectrometers at the Federal Institute for Geosciences and Natural Resources, Hannover/Germany. Instrumental precision of the results was controlled with certified reference materials (CRM) (i.e., BCR, Community Bureau of Reference, Brussels). The precision for major elements was gen-

30 erally better than $\pm 0.5$ % and better than 5 % for trace elements. To decipher a possible secondary alteration of the geochemical composition, the values were expressed as enrichment relative to the detrital background, or average shale, as expressed by Wedepohl (1971); Wedepohl et al. (1991). Accordingly, it is calculated as the element/Al ratio of the sample divided by the element/Al ratio of average shale. Hence, element/Al ratio and enrichment factor are covarying, with a factor > 1 indicating enrichment and < 1 indicating depletion relative to the detrital background.

Biogenic opal was determined photometrically after wet alkaline extraction of biogenic silica (BSi) using a modification of the DeMaster method (DeMaster, 1981). About 30 g dry sediment per sample was digested in 40 mL of 1 % sodium carbonate solution (Na$_2$CO$_3$) in a shaking bath at $85\,°$C. After treatment with 0.021 M HCl, the neutralized supernatant was analyzed after 3, 4 and 5 hours and the amount of BSi was estimated from the linear intercept through the time course aliquots. Slope correction was used to prevent an overestimation of BSi by dissolution of clay minerals at low BSi concentrations (Conley, 1998). Biogenic opal was then determined by multiplying the BSi concentrations with a factor of 2.4. Duplicate measurements revealed a mean standard deviation of 0.13 %.

**3.5 Spectral analyses**

Spectral analyses on the proxy records from planktic foraminiferal transfer functions, Mg/Ca-SST and OMZ intensity with the multi-taper method (MTM; Mann and Lees, 1996) were computed with the SpectraWorks software kSpectra©ver. 3.4.5 and a red noise null hypothesis (Ghil et al., 2002) using the default setting of $p = 2$ and $K = 3$ tapers. A Gaussian band-pass filter was used to reveal the signature of the dominant cycles in the data. After resampling the time series to the average sampling rate, filtering was carried out using the software program AnalySeries ver. 2.0.8 (Paillard et al., 1996). A cross wavelet transform (XWT; Grindsted et al., 2004) of both temperature proxy time series was calculated with the biwavelet package ver. 0.17.10 for $R$. MTM and XWT analyses were conducted on trend-removed time series interpolated to regular average sample spacings using piecewise cubic polynomial interpolation (function 'pchip' of the signal package ver. 0.7-6 for $R$). To estimate a linear relationship of the low-frequency signals between the differently spaced time series, a new common time axis was produced where signals were consecutively averaged into 60-year long bins with a 20-year overlap.

**4 Results**

**4.1 Age control**

The 1–2.5 cm sample spacing yielded an average temporal sampling distance of ~19 years over the entire interval.  The sample spacing is fluctuating between $< 10$ years in the upper 50 cm of the core  and close to 50 years in the lowermost 15 cm of the core. The sharp lithofacies change above the studied interval at 56 cm core depth is marked by a sedimentation hiatus of ~3600 years (Table 1). Based on the accumulation rates above and below the unconformity, this corresponds to a thickness of the missing sedimentary sequence of ~1.5 m. The diatomaceous ooze above the unconformity at 56 cm core depth of SL163 showed an excursive increase of the sedimentary water content of $< 17$ %. One possible reason for  the sedimentation hiatus might thus be gravitational instability due to the high water content of the organic-rich diatomaceous  silty clay deposited

 at the steeply inclined northern Oman margin.  However, further discussion of this issue is beyond the scope of this study and will be presented elsewhere.

We observed two samples where the age-depth relationship is reversed within the lower half of the core (Figure 3). However, the maximum age deviation is lower than the $2\sigma$ probability of both dating points, enabling to fit a smooth spline model with continuous deposition rates and continuously increasing ages. As we did not find any indications of interruptions or other inconsistencies of the sedimentation within the studied interval, we are confident that our age model represents a continuous deposition. Instead, we assume the offset being either the result of changing water mass ages due to changing upwelling intensities, as it was shown for the Peruvian upwelling (Fontugne et al., 2004), or due to mixing by bioturbation.

[revised manuscript text omitted]
. Moreover, the most noticable events of weak ISM conditions are in concordance with advection of North Atlantic drift ice from Bond et al. (2001) (Figure 7e), centerd at ~5.9 (Bond-event 4), ~7.4 (Bond-event 5) and ~8.2 ka B.P. The observation of a contemporaneous occurence of North Atlantic cold spells and weak Asian monsoon was previously found in speleothem (Dykoski et al., 2005; Fleitmann et al., 2007; Liu et al., 2013), lake (Menzel et al., 2014) and other marine records (Gupta et al., 2003). Thus, our finding is further evidence corroborating the presence of an atmopheric teleconnection linking North Atlantic climate to summer monsoonal conditions in tropical Asia during the early- to mid Holocene. However, in order to significantly asses millennial-scale cycles in the upwelling SST off the Oman margin, further studies are needed, as the time span ~2.5 ka covered by our record is likely too short.

MTM analysis revealed statistically significant periodicities of assemblage-based summer SST at ~1300 and 75–95 years per cycle (Figure 8a). Mg/Ca-based upwelling SST are modulated on frequencies at 110–130, 80–90 and ~40 years (Figure 8b). The longer ~110–130 year cycle was previously found in a number of records from the Asian monsoon realm (Berger

and von Rad, 2002; Dykoski et al., 2005; Gupta et al., 2005) and is close to the 132-year  sub-harmonic of the Hale cycle (Attolini et al., 1990) previously identified to be modulating the Oman upwelling system during the Holocene (Gupta et al., 2005). The ~80–90-year cycle has been observed in several studies of ISM variability (Neff et al., 2001; Fleitmann et al., 2003; Dykoski et al., 2005; Gupta et al., 2005) and was interpreted to be most likely influenced by the 88-

5  year solar Gleissberg cycle.  Turner et al. (2016) found, however, that periodicities within the range of the Gleissberg cycle are also common in random-walk simulations and could be a statistical artefact from the sampling resolution and the age model applied. In fact, the overall error of the reservoir correction of ±31 years is close to the observed periodicities, which might give an indication why we found the Gleissberg cycle but not the longer-period (~200–210 years) de Vries solar cycle. However, Duan et al. (2014) showed with a speleothem record from Dongge cave (China), that coherency of East Asian

10  monsoon precipitation and the solar Gleissberg cycle is persistent over the last 4 ka, whereas coherency with the de Vries cycle was only observed over a short 1 ka long period. Other Holocene Asian monsoon records also showed periodicities similar to the Gleissberg cycle, while lacking the longer de Vries cycle (e.g., Wang et al., 1999; Sagawa et al., 2014). It could thus be hypothesized that the de Vries cycle is not detectable in our record, due to a transient link of the solar-monsoon relationship, potentially suppressed by changes in the ocean-atmosphere system in the ENSO domain (Berkelhammer et al., 2010). We

15  therefore tested a possible solar component on the decadal-scale forcing of our SST records by evaluating the coherence of both time series with the record of reconstructed sunspot numbers (Solanki et al., 2004). The coherence pattern reveals, that both SST records and sunspot numbers are coherent on a wide range of periodicities (630, 190–230, 160, 110–130, 80–90, ~70, ~50 and ~40- years per cycle,  Figure 8d and e). The Mg/Ca-based SST record shows strongest coherency with cycles of the sunspot record on both, the ~88-year Gleissberg and the ~190–230-year de Vries periodicities. This observation

20  further strengthens the hypothesis, that ISM variability is not only controlled by orbital-scale insolation forcing, indicated by the long-term trend of warming temperatures and decreasing ISM intensity, but also by solar forcing.

To illustrate the signature of the observed cycles in the data, band-pass filter outputs of both SST records are shown in Figure 9. Apparently, amplitude modulations of the filter outputs of both SST time series run largely synchronous.

25  ~~at 30°N indicates that phases of largest amplitude fluctuations of summer SST are apparently lagging solar irradiance by ~200 years~~ On the ~132-year bandwidth, amplitude modulations of the Mg/Ca-based upwelling SST and TSI are offset by a lag of ~400 years. A direct forcing on this bandwidth seems unlikely, as modelling studies indicate a response of surface temperatures to solar forcing within less than 50 years (Seidenglanz et al., 2012). However, ~~regardless of the exact correlation of both signals, amplitude modulation of upwelling SST in the ~85- and ~120~~ within the limits of our age model, comparison of the amplitude

30  variations of the filter outputs from both SST time series and the total solar irradiance (TSI) record of Steinhilber et al. (2012), filtered on the ~85-year  Gleissberg bandwidth, apparently indicates synchrony of our SST records and TSI with maximum amplitude modulations of all three records centered at ~7.4 ka B.P. Our new high-resolution record of summer/upwelling SST therefore provides further strong evidence that solar forcing was persistently modulating ISM variability during the early- to mid Holocene after the summer insolation maximum,

previously found from speleothem (Neff et al., 2001; Burns et al., 2002; Fleitmann et al., 2003; Dykoski et al., 2005; Wang et al., 2005; Du marine records (von Rad et al., 1999; Wang et al., 1999; Gupta et al., 2005).

[revised manuscript text omitted]

Emerson, S. R. and Huested, S. S.: Ocean anoxia and the concentrations of molybdenum and vanadium in seawater, Marine Chemistry, 34,

35   177–196, 1991.

Emery, W. J. and Meincke, J.: Global Water Masses - Summary and Review, Oceanologica Acta, 9, 383–391, 1986.

Fairbanks, R. G., Sverdlove, M., Free, R., Wiebe, P. H., and Bé, A.: Vertical-Distribution and Isotopic Fractionation of Living Planktonic-Foraminifera From the Panama Basin, Nature, 298, 841–844, 1982.

Farías, L., Fernández, C., Faúndez, J., et al.: Chemolithoautotrophic production mediating the cycling of the greenhouse gases $N_2O$ and $CH_4$ in an upwelling ecosystem, Biogeosciences, 6, 3053–3069, 2009.

Findlater, J.: A major low-level air current near the Indian Ocean during the northern summer, Quarterly Journal of the Royal Meteorological Society, 95, 362–380, 1969.

5     Fleitmann, D., Burns, S. J., Mudelsee, M., Neff, U., Kramers, J., Mangini, A., and Matter, A.: Holocene forcing of the Indian monsoon recorded in a stalagmite from Southern Oman, Science, 300, 1737–1739, 2003.

Fleitmann, D., Burns, S. J., Neff, U., Mudelsee, M., Mangini, A., and Matter, A.: Palaeoclimatic interpretation of high-resolution oxygen isotope profiles derived from annually laminated speleothems from Southern Oman, Quaternary Science Reviews, 23, 935–945, 2004.

Fleitmann, D., Burns, S. J., Mangini, A., and Mudelsee, M.: Holocene ITCZ and Indian monsoon dynamics recorded in stalagmites from
10     Oman and Yemen (Socotra), Quaternary Science Reviews, 26, 170–188, 2007.

Fontugne, M., Carre, M., Bentaleb, I., Julien, M., and Lavallee, D.: Radiocarbon reservoir age variations in the south Peruvian upwelling during the Holocene., Radiocarbon, 46, 531–537, 2004.

Friedrich, O., Schiebel, R., Wilson, P. A., and Weldeab, S.: Influence of test size, water depth, and ecology on Mg/Ca, Sr/Ca, $\delta$ 18 O and $\delta$ 13 C in nine modern species of planktic foraminifers, Earth an Planetary Science Letters, 319-320, 133–145, 2012.

[revised manuscript text omitted]

Milliman, J. D., Troy, P. J., Balch, W. M., and Adams, A. K.: Biologically mediated dissolution of calcium carbonate above the chemical lysocline?, Deep Sea Research Part I: Oceanographic Research Papers, 46, 1653–1669, 1999.

Mohan, R., Verma, K., Mergulhao, L. P., Sinha, D. K., Shanvas, S., and Guptha, M. V. S.: Seasonal variation of pteropods from the Western Arabian Sea sediment trap, Geo-Marine Letters, 26, 265–273, 2006.

Mohtadi, M., Oppo, D. W., Lückge, A., DePol-Holz, R., Steinke, S., Groeneveld, J., Hemme, N., and Hebbeln, D.: Reconstructing the thermal structure of the upper ocean: Insights from planktic foraminifera shell chemistry and alkenones in modern sediments of the tropical eastern Indian Ocean, Paleoceanography, 26, PA3219, 2011.

Mohtadi, M., Prange, M., Oppo, D. W., De Pol-Holz, R., Merkel, U., Zhang, X., Steinke, S., and Lückge, A.: North Atlantic forcing of tropical Indian Ocean climate, Nature, 509, 76–80, 2014.

Morse, J. W., Mucci, A., and Millero, F. J.: The solubility of calcite and aragonite in seawater at various salinities, temperatures and atmosphere total pressure, Geochimica et Cosmochimica Acta, 44, 85–94, 1980.

Munz, P. M., Siccha, M., Lückge, A., Böll, A., Kucera, M., and Schulz, H.: Decadal-resolution record of winter monsoon intensity over the last two millennia from planktic foraminiferal assemblages in the northeastern Arabian Sea, The Holocene, 25, 1756–1771, 2015.

Munz, P. M., Lückge, A., Siccha, M., Böll, A., Forke, S., Kucera, M., and Schulz, H.: The Indian winter monsoon and its response to external forcing over the last two and a half centuries, Climate Dynamics, doi:10.1007/s00382-016-3403-1, 2016.

Murtugudde, R., Seager, R., and Thoppil, P.: Arabian Sea response to monsoon variations, Paleoceanography, 22, PA4217, 2007.

[revised manuscript text omitted]

ter Braak, C. J. F. and Juggins, S.: Weighted averaging partial least squares regression (WA-PLS): An improved method for reconstructing environmental variables from species assemblages, Hydrobiologia, 269–270, 485–502, 1993.

Thamban, M., Kawahata, H., and Rao, V. P.: Indian summer monsoon variability during the holocene as recorded in sediments of the Arabian Sea: Timing and implications, Journal of oceanography, 63, 1009–1020, 2007.

Tomczak, M. and Godfrey, J. S.: Regional Oceanography, Pergamon, Oxford, U.K., 1994.

Tribovillard, N., Algeo, T. J., Lyons, T., and Riboulleau, A.: Trace metals as paleoredox and paleoproductivity proxies: An update, Chemical Geology, 232, 12–32, 2006.

Turner, T. E., Swindles, G. T., Charman, D. J., Langdon, P. G., Morris, P. J., Booth, R. K., Parry, L. E., and Nichols, J. E.: Solar cycles or random processes? Evaluating solar variability in Holocene climate records., Scientific Reports, 6, 23 961, 2016.

von Rad, U., Schaaf, M., Michels, K. H., Schulz, H., Berger, W. H., and Sirocko, F.: A 5000-yr Record of Climate Change in Varved Sediments from the Oxygen Minimum Zone off Pakistan, Northeastern Arabian Sea, Quaternary Research, 51, 39, 1999.

Wang, L., Sarnthein, M., Erlenkeuser, H., Grimalt, J., Grootes, P., Heilig, S., Ivanova, E., Kienast, M., Pelejero, C., and Pflaumann, U.: East Asian monsoon climate during the Late Pleistocene: high-resolution sediment records from the South China Sea, Mar. Geol., 156, 245, 1999.

Wang, Y., Cheng, H., Edwards, R. L., He, Y., Kong, X., and An, Z.: The Holocene Asian monsoon: links to solar changes and North Atlantic climate, Science, 308, 854–857, 2005.

[revised manuscript text omitted]

**Figure 7.** Comparison of SST time series from **(a)** the Mg/Ca ratio of *G. bulloides* and **(b)** assemblage-based transfer functions with **(c)** $\delta^{18}O$ ratio (‰) of *G. ruber* from core 63KA (Staubwasser et al., 2003) and **(d)** the colour reflectance (L*) of core SO130-289KL (Deplazes et al., 2013) from the Pakistan margin indicative of precipitation and productivity associated with ISM intensity. **(e)** Hematite stained grains (HSG %) indicate ice rafting events and cold spells in the North Atlantic (Bond et al., 2001). Bond events 4 (~5.9 ka B.P.) and 5 (~7.4 ka B.P.) are indicated.

[Figure]

**Figure 8.** Results of the MTM spectral analysis show the frequency distribution and identified significant periodicities of the **(a)** assemblage-based SST record, **(b)** the SST record using Mg/Ca ratios of *G. bulloides* and **(c)** the Mn/Al ratio as a proxy for OMZ conditions. Black lines indicate 90 % and 95 % significance levels against red noise. Cross-spectral coherence of both SST records (**d** and **e**) with the record of sunspot numbers (Solanki et al., 2004) indicates that both records significantly covary on a wide range of frequencies. All time series were trend-removed prior to the analyses.

[Figure]

**Figure 9.** Comparison of filtered assemblage-based SST (red line, upper panel) and *G. bulloides* Mg/Ca SST (blue lines, lower panel) with  filtered reconstructed total solar irradiance (TSI) from Steinhilber et al. (2012) (grey lines), 88 years per cycle in the upper panel and 132 years per cycle in the lower panel. Filtered components were extracted using a Gaussian band-pass filter with a central frequency of 0.0118 cycles year$^{-1}$ (~85 years cycle) and 0.0083 cycles year$^{-1}$ (~120-years cycle), respectively, and a bandwidth of 0.001 cycles year$^{-1}$  using the  program AnalySeries 2.0.8 (Paillard et al., 1996).

[Figure]

**Figure 10.** Continuous wavelet transform of **(a)** assemblage-based summer SST and **(b)** Mg/Ca-based upwelling SST. The 10 % significance level against red noise is shown as black contour lines. **(c)**Cross wavelet transform (XWT) of the trend-removed Mg/Ca-based and assemblage-based SST time series. The 5 % significance level is shown as thick contours. Relative phasing of both records is indicated by arrows. Arrows within the significant intervals pointing mostly upwards, which indicates a consistent phase-shift of both records by 90° or 270°, respectively, of assemblage-based SST relative to Mg/Ca-based SST. The cone of influence, where edge artefacts might be introduced, is indicated by hatching and white shading, respectively.

**Table 1.** Samples for radiocarbon dating with the organic compound used, dating results and calibrated ages using the MARINE13 calibration curve (Reimer et al., 2013) and a reservoir correction of $\Delta R = 231 \pm 31$ years. Analyses were conducted at Beta Analytics in Miami/FL, U.S.A. (Beta) and the Leibniz Laboratory in Kiel, Germany (KIA).

| Lab Code | Core | Sample Type | Depth (cm) | Conventional $^{14}$C age (years) | Error $\pm$ | $\delta^{13}$C (‰) | cal. min (95%) | cal. max (95%) |
|---|---|---|---|---|---|---|---|---|
| Beta342813 | MC681 | mixed PF | 1 | 650 | 30 | –2.4 | | |
| Beta342812 | MC681 | bulk org. | 1 | 630 | 30 | –20.4 | | |
| Beta346603 | SL163 | bulk org. | 52.5 | 2100 | 30 | –19.4 | 1329 | 1505 |
| Beta346604 | SL163 | bulk org. | 58.5 | 5740 | 30 | –19.3 | 5788 | 5973 |
| KIA47119 | SL163 | *N. dutertrei* | 77.75 | 5760 | 30 | +0.24 | 5841 | 5987 |
| Beta319751 | SL163 | *N. dutertrei* | 141.25 | 5990 | 30 | –0.5 | 6095 | 6266 |
| Beta319752 | SL163 | *N. dutertrei* | 193.75 | 6350 | 40 | +0.5 | 6436 | 6646 |
| KIA47120 | SL163 | *N. dutertrei* | 252.75 | 6715 | 35 | +0.48 | 6878 | 7114 |
| Beta319753 | SL163 | *N. dutertrei* | 278.75 | 6670 | 40 | +0.8 | 6790 | 7043 |
| Beta319754 | SL163 | *N. dutertrei* | 295.0 | 7030 | 40 | +0.3 | 7246 | 7407 |
| Beta319755 | SL163 | *N. dutertrei* | 326.25 | 6990 | 40 | +0.4 | 7193 | 7386 |
| KIA47121 | SL163 | *N. dutertrei* | 353.75 | 7420 | 40 | +1.59 | 7568 | 7737 |
| KIA47122 | SL163 | *N. dutertrei* | 392.5 | 8090 | 40 | +1.42 | 8211 | 8396 |
| Beta319756 | SL163 | *N. dutertrei* | 395.0 | 8090 | 40 | +0.8 | 8211 | 8396 |
| Beta342816 | SL163 | *N. dutertrei* | 417.5 | 8500 | 40 | –0.4 | 8647 | 8957 |

mixed PF=mixed planktic foraminifera; bulk org.=bulk organic fraction

**Table 2.** Results of the redundancy analysis of the adjusted calibration dataset. Highest correlation with RDA1 axis and highest explanation of the PF assemblage variance is shown by temperature (bold numbers).

| | RDA1 | RDA2 | RDA3 | Captured variance of species data | **Proportion (%)** |
|---|---|---|---|---|---|
| Eigenvalue | 1.763 | 0.185 | 0.097 | | |
| Cumulative proportion of variance explained (%) | 35.9 | 39.6 | 41.6 | | |
| **Correlation** | | | | | |
| Temperature | **–0.93** | –0.07 | 0.15 | 1.55 | **31.5** |
| Salinity | –0.87 | 0.47 | –0.05 | 0.24 | 5.0 |
| Eppley-VGPM | 0.33 | 0.75 | 0.26 | 0.19 | 4.0 |
| uCbPM | –0.02 | –0.65 | 0.32 | 0.07 | 1.5 |
| Chlorophyll $\alpha$ | 0.39 | 0.36 | –0.18 | 0.05 | 1.0 |

VGPM=Vertical Generalized Production Models; uCbPM=updated Carbon-based Production Model.

**Table 3.** Cross-validated root mean squared error of prediction (RMSEP) as absolute values and relative to the range of the target variable, as well as the coefficient of determination. The results of the significance analysis after Telford and Birks (2011) from both transfer function methods are given, $p$ indicates the statistical significance against the null distribution of 999 random reconstructions.

| | RMSEP | RMSEP (% of target range) | $R^2$ | SST $p =$ | Eppley-VGPM $p =$ |
|---|---|---|---|---|---|
| Imbrie-Kipp | 0.92 | 15.72 | 0.67 | 0.015 | 0.758 |
| WA-PLS | 0.95 | 16.20 | 0.65 | 0.007 | 0.160 |

WA-PLS=weighted averaging partial least squares method.